

# A biogenic CO$_2$ flux adjustment scheme for the mitigation of large-scale biases in global atmospheric CO$_2$ analyses and forecasts

A. Agustí-Panareda[1], S. Massart[1], F. Chevallier[2], G. Balsamo[1], S. Boussetta[1], E. Dutra[1], and A. Beljaars[1]

[1]European Centre for Medium-Range Weather Forecasts, Reading, UK
[2]Laboratoire des Sciences du Climat et l'Environnement, Gif-sur-Yvette, France

Received: 7 December 2015 – Accepted: 10 December 2015 – Published: 19 January 2016

Correspondence to: A. Agustí-Panareda (anna.agusti-panareda@ecmwf.int)

Published by Copernicus Publications on behalf of the European Geosciences Union.

**ACPD**

doi:10.5194/acp-2015-987

**Biogenic flux adjustment scheme for CO$_2$ analysis and forecasting system**

A. Agustí-Panareda et al.



**ACPD**

doi:10.5194/acp-2015-987

**Biogenic flux adjustment scheme for CO$_2$ analysis and forecasting system**

A. Agustí-Panareda et al.

## Abstract

Forecasting atmospheric CO$_2$ daily at the global scale with a good accuracy like it is done for the weather is a challenging task. However, it is also one of the key areas of development to bridge the gaps between weather, air quality and climate models. The challenge stems from the fact that atmospheric CO$_2$ is largely controlled by the CO$_2$ fluxes at the surface, which are difficult to constrain with observations. In particular, the biogenic fluxes simulated by land surface models show skill in detecting synoptic and regional-scale disturbances up to sub-seasonal time-scales, but they are subject to large seasonal and annual budget errors at global scale, usually requiring a posteriori calibration. This paper presents a scheme to diagnose and mitigate model errors associated with biogenic fluxes within an atmospheric CO$_2$ forecasting system. The scheme is an adaptive calibration referred to as Biogenic Flux Adjustment Scheme (BFAS) and it can be applied automatically in real time throughout the forecast. The BFAS method improves the continental budget of CO$_2$ fluxes in the model by combining information from three sources: (1) retrospective fluxes estimated by a global flux inversion system, (2) land-use information, (3) simulated fluxes from the model. The method is shown to produce enhanced skill in the daily CO$_2$ 10-day forecasts without requiring continuous manual intervention. Therefore, it is particularly suitable for near-real-time CO$_2$ analysis and forecasting systems.

## 1 Introduction

Earth-observing strategies focusing on carbon cycle systematic monitoring from satellites and in situ networks (Ciais et al., 2014; Denning et al., 2005) are leading to an increasing number of near-real-time observations available to systems such as those developed in the framework of the European Union Copernicus Atmosphere Monitoring Service (CAMS). CAMS uses the Numerical Weather Prediction (NWP) Integrated Forecasting system (IFS) of the European Centre for Medium range Weather Forecasts

Title Page

| Abstract | Introduction |
| Conclusions | References |
| Tables | Figures |

◄◄ ►►

◄ ►

Back | Close

Full Screen / Esc

Interactive Discussion

(ECMWF) to produce near-real-time global atmospheric composition analysis and forecasts, including $CO_2$ (Agustí-Panareda et al., 2014) along with other environmental and climate relevant tracers (Flemming et al., 2009; Morcrette et al., 2009; Massart et al., 2014).

The present monitoring of global atmospheric $CO_2$ relies on observations of atmospheric $CO_2$ from satellites – e.g. Greenhouse Gases Observing Satellite (GOSAT, www.gosat.nies.go.jp); Orbiting Carbon Observatory 2 (OCO-2, oco.jpl.nasa.gov) – and in situ networks – e.g. National Oceanic and Atmospheric Administration Earth System Research Laboratory (NOAA/ESRL, www.esrl.noaa.gov/gmd); Integrated Car-
bon Observation System (ICOS, icos-atc.lsce.ipsl.fr); Environment Canada (www.ec.gc.ca/mges-ghgm) – which are assimilated by global tracer transport models to infer changes in atmospheric $CO_2$ (e.g. Massart et al., 2015) or by flux inversion systems (e.g. Peylin et al., 2013) to estimate the large-scale surface fluxes of $CO_2$.

The current CAMS $CO_2$ analysis is produced by assimilating $CO_2$ data retrieved from
15 GOSAT by the University of Bremen (Heymann et al., 2015), as well as all the meteorological data that is routinely assimilated in the operational meteorological analysis at ECMWF. Massart et al. (2015) have shown that the atmospheric data assimilation system alone cannot completely remove the biases in the background atmospheric $CO_2$ associated with the accumulation of errors in the $CO_2$ fluxes from the model. This
happens because currently the $CO_2$ surface fluxes in the IFS data assimilation system cannot be constrained by observations. In this paper, we present a method to reduce the atmospheric $CO_2$ model biases by adjusting the $CO_2$ surface fluxes in a near-real-time $CO_2$ analysis/forecasting system, such as the one used by CAMS at ECMWF.

Many different methods already exists to adjust $CO_2$ fluxes by using observations
of atmospheric $CO_2$ within flux inversion systems (Rödenbeck et al., 2003; Gurney et al., 2003; Peters et al., 2007). However, these are not all suitable for the CAMS real-time monitoring system. Flux inversion systems adjust the fluxes by either inferring the model parameters in Carbon Cycle Data Assimilation Systems also known as CCDAS (Rayner et al., 2005; Scholze et al., 2007; Rayner et al., 2011), or the fluxes

Discussion Paper | Discussion Paper | Discussion Paper | Discussion Paper |

**[ACPD](doi:10.5194/acp-2015-987)**

doi:10.5194/acp-2015-987

**Biogenic flux adjustment scheme for CO₂ analysis and forecasting system**

A. Agustí-Panareda et al.

themselves (Houweling et al., 2015). CCDAS has the advantage of working in prognostic mode once the model parameters have been optimised. Nevertheless, it can also be prone to aliasing information to the wrong model parameter when the processes that contribute to the variability of atmospheric $CO_2$ are not properly represented in the model or missing altogether. Estimating directly the $CO_2$ fluxes does not rely on the accurate representation of complex/unknown processes in the $CO_2$ flux model, but the resulting optimised fluxes do not have predictive skill. Both approaches generally use long data assimilation windows of several weeks to years in order to be able to constrain the global mass of $CO_2$ by relying mainly on high quality in situ observations which are relatively sparse in time and space. This general requirement for long assimilation windows is incompatible with the current NWP framework (e.g. a 12-h window is currently used in the IFS). In addition to that, the $CO_2$ observations from flask and most in situ stations used by these flux inversion systems are not available in near-real time.

Considering all the aspects mentioned above, a Biogenic Flux Adjustment Scheme (hereafter called BFAS) suitable for the NWP framework is proposed which aims to combine the best characteristics of both flux inversion approaches. Namely, the mass constraint from the optimised fluxes is used to correct the biases of the modelled $CO_2$ fluxes while keeping the predictive skill of the modelled fluxes at synoptic scales. The main objective of BFAS is to reduce the large-scale biases of the background atmospheric $CO_2$. This should improve the representation of the atmospheric $CO_2$ large-scale gradients, and thereby also lead to a better forecast of atmospheric $CO_2$ synoptic variability.

The details of the flux adjustment scheme are provided in Sect. 2. Section 3 describes the IFS experiments done to test the impact of BFAS on the atmospheric $CO_2$ forecast. From the experiments, different aspects of the flux adjustment can be monitored (i.e. the scaling factors and the resulting budget) as shown in Sect. 4. The resulting atmospheric $CO_2$ forecast fit to observations after applying BFAS is presented in Sect. 5. The potential use of BFAS for model development and the possibility of in-

Discussion Paper | Discussion Paper | Discussion Paper | Discussion Paper |

**[ACPD](doi:10.5194/acp-2015-987)**

doi:10.5194/acp-2015-987

**Biogenic flux adjustment scheme for $CO_2$ analysis and forecasting system**

A. Agustí-Panareda et al.

cluding BFAS in the data assimilation system are discussed in Sect. 6. Finally, Sect. 7 gives a summary of the flux adjustment achievements and possible developments for the future.

## 2 Methodology

The flux adjustment scheme aims at reducing the large-scale biases in the background atmospheric $CO_2$ of the current CAMS forecasting system. Agustí-Panareda et al. (2014) documented the configuration of the $CO_2$ forecasting system and showed that the large biases in atmospheric $CO_2$ are consistent with errors associated with the budget of $CO_2$ surface fluxes. Optimised fluxes from flux inversion systems constitute the best available estimate of the $CO_2$ fluxes given the observed variations of $CO_2$ in the atmosphere at global scales. Thus, they can provide a reference benchmark for the modelled fluxes. The large-scale biases in the $CO_2$ fluxes can be diagnosed by computing the budget (i.e. integrated) differences between modelled fluxes and optimised fluxes over continental and supra-synoptic spatial and temporal scales ($\geq 1000$ km, 10 days). Working with budgets over scales beyond the synoptic scale allows the detection of large-scale biases without interfering with the synoptic skill of the model.

It is important to note that there are uncertainties and limitations that should be considered when using optimised fluxes. Optimised fluxes are computed with flux inversion systems at low resolutions ($\sim$ hundreds of km) compared to the NWP resolution used for the $CO_2$ forecasts ($\sim$ tens of km), and they are most reliable at continental and supra-synoptic scales. Moreover, they have the limitation of not being available in near-real time, unlike the meteorological observations or $CO_2$ satellite retrievals (Massart et al., 2015). Because of that, a climatology of the optimised fluxes has to be used as a reference. Finally, optimised fluxes only provide information on the total $CO_2$ flux because flux inversion systems are not able to attribute the $CO_2$ variability to the different processes controlling the fluxes, such as vegetation, anthropogenic sources and fires.

**ACPD**

doi:10.5194/acp-2015-987

**Biogenic flux adjustment scheme for CO2 analysis and forecasting system**

A. Agustí-Panareda et al.

The $CO_2$ forecast evaluation by Agustí-Panareda et al. (2014) showed that the Net Ecosystem Exchange (NEE) modelled by the CTESSEL carbon model (Boussetta et al., 2013) within the IFS is the main responsible for the large global biases in the atmospheric $CO_2$ seasonal cycle. Generally, the land $CO_2$ fluxes from vegetation and soils in models are associated with high uncertainty (Le Quéré et al., 2015). For this reason, the Global Carbon Project provides the $CO_2$ budget from land vegetation – also known as the land sink – as a residual to close the carbon budget (see www.globalcarbonproject.org/carbonbudget). Following the land sink residual approach, the optimised NEE can be computed as the residual of optimised fluxes by subtracting the other prescribed fluxes. A set of 10-day mean budgets of this residual NEE from optimised fluxes is then computed daily for different regions and vegetation types over a period of 10 years to build the NEE climatology that can be used as a reference. In order to account for the inter-annual variability of NEE, the reference climatology is also adjusted with an inter-annual variability factor obtained from the model.

The flux adjustment scheme essentially estimates the bias of the modelled NEE budget with respect to the reference NEE budget for each region and vegetation type as a scaling factor $\alpha$:

$$\alpha = \frac{f^{O}}{f^{M}} \tag{1}$$

where $f$ is the 10-day mean NEE budget computed daily over a specific vegetation type and region, $f^{O}$ is the reference budget based on the MACC-13R1 optimised fluxes (Chevallier et al., 2010), and $f^{M}$ is the budget of the modelled fluxes. Figure 1 shows how the BFAS scheme interacts with the model to produce the flux-corrected atmospheric $CO_2$ forecast. First of all, the uncorrected NEE fluxes from the model are retrieved. Then their budget is compared with the budget of the NEE climatology from the optimised fluxes adjusted with the NEE anomaly from the model. The scheme produces maps with scaling factors of the biogenic fluxes before the forecast run. Subsequently,

**ACPD**

doi:10.5194/acp-2015-987

**Biogenic flux adjustment scheme for $CO_2$ analysis and forecasting system**

A. Agustí-Panareda et al.

these maps are then used to scale the forecast of NEE. There are three major building blocks required for the computation of these scaling factors:

– The computation of the NEE budget using temporal and spatial aggregation criteria (e.g. 10 days, vegetation types, different regions).

– A reference NEE dataset used to diagnose the model biases (e.g. optimised fluxes from global flux inversion systems such as the MACC-13R1 dataset from Chevallier et al. (2010)).

– The partition of the NEE adjustment into the two modelled ecosystem fluxes that make up the NEE flux: i.e. Gross Primary Production (GPP) associated with photosynthesis and ecosystem respiration ($R_{eco}$) documented by Boussetta et al. (2013).

These different aspects are discussed in further detail below in Sect. 2.1 to 2.3.

## 2.1 Computation of NEE budget

The biases of the NEE fluxes that we aim to correct are partly linked to model parameter errors that depend on vegetation type and to errors in the meteorological/vegetation state which are region-dependent (e.g. radiation, LAI, temperature and precipitation). In addition to that, the global optimised fluxes used as reference do not currently have a strong constraint from observations at small spatial and temporal scales due to the sparse observing network of atmospheric $CO_2$. Therefore, the NEE biases are not diagnosed at the model grid-point scale, but as biases in the NEE budget over continental regions for different vegetation types and over a period of 10 days. The 10-day regional budget provides an indicator on the large-scale biases. Moreover, 10 days is a period that can be used in the current framework of the CAMS global atmospheric $CO_2$ forecasting system. Figure 2 shows how the uncorrected NEE from the past forecasts can be combined to compute the 10-day mean budget before each new forecast. The 1-day forecasts initialised from the previous seven days are used together with the last

Discussion Paper | Discussion Paper | Discussion Paper | Discussion Paper

**ACPD**

doi:10.5194/acp-2015-987

**Biogenic flux adjustment scheme for CO$_2$ analysis and forecasting system**

A. Agustí-Panareda et al.

**[ACPD](https://www.atmos-chem-phys.net/)**

doi:10.5194/acp-2015-987

**Biogenic flux adjustment scheme for CO$_2$ analysis and forecasting system**

A. Agustí-Panareda et al.

3-day forecast available in order to create a 10-day window around the initial date of the new forecast. This 10-day time window is slightly shifted to the past because otherwise forecasts longer than 3-days would be required to compute the budget while errors in the meteorology affecting the fluxes grow with forecast lead time. Chevallier and Kelly (2002) found that forecast errors associated with the location of extra-tropical weather systems affecting the cloud cover and temperature gradients – which in turn will affect the NEE errors – are very small at day 1. These errors continue to be small up to day 3, but they can grow rapidly with forecast lead time (see Haiden et al., 2015, for details on the IFS forecast error evaluation). The different regions have been selected according to latitudinal band characterised by seasonal cycle (northern hemisphere, tropics and southern hemisphere), continental region and vegetation type.

In the IFS the vegetation types follow the BATS classification (Dickinson et al., 1986), which is widely used in meteorological and climate models. The vegetation classification is designed to distinguish between roughness lengths for the computation of the momentum, heat and moisture transfer coefficients in the modelling of the fluxes from surface to atmosphere. However, the BATS vegetation types are not always suitable for the modelling of the CO$_2$ fluxes. For example, the interrupted forest type which constitutes around 25 % of the high vegetation cover encompasses many different types of vegetation, including Tropical Savanna and a combination of remnants of forest or open woods lands with field complexes. This could be an important source of error in some regions. For this reason, BFAS allows the introduction of new vegetation types for diagnosing the NEE biases. Tropical Savanna which covers large areas in the tropical region has been added as a subtype of the interrupted forest vegetation type by using the Olson Global Ecosystem classification (Olson, 1994a, b, edc2.usgs.gov/glcc/globdoc2_0.php).

Figure 3 shows the distribution of the dominant vegetation types used in BFAS. Land cover maps from GLCC version 1 (edc2.usgs.gov/glcc/glcc.php) are used to compute the land cover of the dominant high and low vegetation types at each grid point. In BFAS, only one dominant vegetation type is used to classify each grid point, and this

must cover more than 50 % of the grid box. Model grid points with less than 50 % vegetation cover are not used. The comparison of the modelled NEE with the optimised NEE fluxes is done by computing 10-day budgets for each of the 16 vegetation types (see Table 1) and 9 different regions (see Fig. 3).

## 2.2 Reference NEE budget

The residual NEE from optimised fluxes provides the reference for the flux adjustment scheme. Currently, there is no operational centre providing $CO_2$ optimised fluxes at global scale in near-real time. We have chosen to use the MACC optimised fluxes (Chevallier et al., 2010) which are delivered around September each year for the previous year. The MACC optimised $CO_2$ fluxes are regularly improved and their high quality has been recently shown by Kulawik et al. (2015). Chevallier (2013) provides an evaluation of the inverted $CO_2$ fluxes for 2010.

The computation of the residual is done by subtracting the prescribed fluxes used in the CAMS $CO_2$ forecast over land from the total optimised flux. The prescribed $CO_2$ fluxes from biomass burning and anthropogenic emissions in the $CO_2$ forecast are not the same as the ones used as prior fluxes in the MACC flux inversion system. Not only they are from different sources, but they are also used at different resolutions. This means that there might be fires represented in one and not the other, or with different emission intensities, as it is the case for anthropogenic hotspots at high versus low resolutions. Thus, in order to avoid the transfer of inconsistencies between the prescribed and prior fluxes into the NEE residual, the regions with very high anthropogenic emissions (larger than $3 \times 10^6 \, \mathrm{g\,C\,m^{-2}\,s^{-1}}$) and fires are filtered out.

A climatology of these reference NEE fluxes is created using the last 10 available years and it is updated every time a new year is available. Thus, allowing for slow decadal variations in the NEE reference. Figure 4 shows a comparison of the optimised flux budget in 2010 and its climatology for the crop vegetation type in North America. The inter-annual variability of the optimised flux budget is depicted by the standard deviation around the 10-year climatology. The reference NEE climatology is then adjusted

Discussion Paper | Discussion Paper | Discussion Paper | Discussion Paper | Discussion Paper |

**ACPD**

doi:10.5194/acp-2015-987

**Biogenic flux adjustment scheme for CO$_2$ analysis and forecasting system**

A. Agustí-Panareda et al.

to account for the inter-annual variability of the land sink fluxes as follows:

$$f^O = f^{Oclim} + \gamma \sigma \left( f^{Oclim} \right), \tag{2}$$

where $f$ is the 10-day NEE budget for a specific region and vegetation type, $f^O$ is the reference budget, $f^{Oclim}$ and $\sigma(f^{Oclim})$ are the climatological mean and standard deviation of the optimised flux budget respectively from 2004 to 2013, and $\gamma$ is the corresponding standardised anomaly of the NEE budget from the model with respect to the same period. $\gamma$ can be positive or negative. It represents the inter-annual variability factor used to adjust the reference climatological NEE budget and it is given by

$$\gamma = \frac{f^M - f^{Mclim}}{\sigma \left( f^{Mclim} \right)} \tag{3}$$

where $f^M$ is the model NEE budget, $f^{Mclim}$ is the climatological mean budget from the model and $\sigma(f^{Mclim})$ is the standard deviation of the model NEE budget denoting the typical amplitude of its inter-annual variability for the same period as the climatology of the optimised flux budget (i.e. 2004 to 2013).

The $\gamma$ inter-annual variability factor is multiplied by the standard deviation of the optimised residual NEE budget – representing the typical amplitude of inter-annual variability – in order to offset the reference climatological NEE budget. In this way, the inter-annual variability of the reference NEE follows the inter-annual variability of the model NEE with the same anomaly sign, while keeping its amplitude constrained by the standard deviation of the optimised flux budget.

The computation of $\gamma$ requires a model climate consistent with the forecast (i.e. same meteorological analysis, same model version and same resolution). Producing a consistent model climate is not a trivial requirement, because both the operational model version and analysis system can change frequently with new updates and new observations, and high resolution forecasts spanning a period of 10 years (i.e. 2004 to

Discussion Paper | Discussion Paper | Discussion Paper | Discussion Paper | Discussion Paper |

**ACPD**

doi:10.5194/acp-2015-987

**Biogenic flux adjustment scheme for CO$_2$ analysis and forecasting system**

A. Agustí-Panareda et al.

2013) are expensive. A feasible solution has been found where the standardised NEE anomaly from the model is computed using the operational Ensemble Prediction System (ENS) forecasts and hindcasts which are part of the ECMWF monthly forecasting system (Vitart et al., 2008; Vitart, 2013, 2014). Every Monday and Thursday the op-

5 erational ENS is not only run for the actual date, but also for the same calendar day of the past 20 years. These hindcasts have the same resolution and model version as the ENS forecasts and they constitute a valuable data set used for the post-processing and calibration of the NWP forecasts from the medium-range (10 days) up to one month lead times (Hagedorn et al., 2012). The ensemble of forecasts is made of 5 members

(10 members since 2015) using perturbed initial conditions (Lang et al., 2015) and stochastic physics in order to represent forecast uncertainty (Palmer et al., 2009).

As the hindcasts are not performed daily, it is not possible to aggregate consecutive 1-day forecasts into a 10-day period to compute a mean budget as shown in Fig. 2. In order to circumvent this, the mean budget is computed by averaging the 1-day forecast

NEE from all the ensemble members available in the hindcasts. This is done for each year from 2004 to 2013 to preserve consistency with the NEE climatology from the optimised fluxes. The model climate $f^{\mathrm{Mclim}}$ given by the 10-year mean budget and its typical inter-annual variability $\sigma\left(f^{\mathrm{Mclim}}\right)$ can then be obtained by calculating the mean value and standard deviation respectively over that period. Similarly, the model budget

$f^{\mathrm{M}}$ is calculated from the NEE ensemble mean of the ENS forecast for the current date using the same number of ensemble members as the ENS hindcasts. The standardised anomaly $\gamma$ is finally obtained by subtracting the 10-year mean budget from the current budget and dividing the anomaly by the standard deviation. Since the hindcasts are available every Monday and Thursday, $\gamma$ is only updated twice a week. These updates

are routinely monitored during the forecast (see Sect. 4).

**[ACPD](doi:10.5194/acp-2015-987)**

doi:10.5194/acp-2015-987

**Biogenic flux adjustment scheme for CO$_2$ analysis and forecasting system**

A. Agustí-Panareda et al.

Discussion Paper | Discussion Paper | Discussion Paper | Discussion Paper | Discussion Paper |

**[ACPD](doi:10.5194/acp-2015-987)**

doi:10.5194/acp-2015-987

**Biogenic flux adjustment scheme for CO$_2$ analysis and forecasting system**

A. Agustí-Panareda et al.

## 2.3 Partition of NEE adjustment

The final stage in the flux adjustment is the attribution of the NEE correction to the different biogenic fluxes in the model. The residual NEE from optimised fluxes only provides information on the total flux from the land ecosystem exchange. While in land vegetation models, NEE is the combination of two opposing fluxes: Gross Primary Production (GPP) and the ecosystem respiration ($R_{eco}$). Given that we have no information on whether the NEE error is associated with the GPP or the $R_{eco}$ fluxes, a strategy has to be defined in order to partition the NEE correction into GPP and $R_{eco}$. The underlying strategy used here is to have the smallest flux adjustment possible. Namely, the scaling factors should be as close to 1 as possible.

The first step is to distinguish between the positive and negative values of the NEE scaling factor ($\alpha$). A positive NEE scaling factor implies the budget of the NEE in the model has the correct sign but the wrong magnitude. In that case, the scaling of the flux will be smallest if the dominant component of NEE is scaled. That is to say, the flux correction will be applied to GPP during the growing season and to $R_{eco}$ during the senescence period. Whereas if the scaling factor is negative – i.e. the modelled NEE has the wrong sign – only the flux with smallest magnitude is corrected (GPP or $R_{eco}$) to ensure the scaling factor of the modelled fluxes is always positive.

The scaling factor $\alpha$ is then converted into a scaling factor for the dominant component of the NEE flux. If the magnitude of GPP is larger than the magnitude of $R_{eco}$, then the scaling factor for GPP and $R_{eco}$ are defined as follows:

$$\alpha_{GPP} = \frac{\alpha NEE - R_{eco}}{GPP}$$
$$\alpha_{R_{eco}} = 1.0 \tag{4}$$

**ACPD**

doi:10.5194/acp-2015-987

**Biogenic flux adjustment scheme for CO$_2$ analysis and forecasting system**

A. Agustí-Panareda et al.

Similarly, if $|R_{\text{eco}}| > |\text{GPP}|$ then

$$\alpha_{\text{GPP}} = 1.0$$
$$\alpha_{R_{\text{eco}}} = \frac{\alpha\text{NEE} - \text{GPP}}{R_{\text{eco}}} \tag{5}$$

This partition the flux adjustment is a modelling choice based on minimum flux adjustment criteria. Other solutions might be possible given additional information on either GPP or $R_{\text{eco}}$ budgets.

The $\alpha_{\text{GPP}}$ and $\alpha_{R_{\text{eco}}}$ factors are computed for each vegetation type and region and then re-mapped as 2-d fields using the dominant vegetation type map in Fig. 3. The resulting maps for $\alpha_{\text{GPP}}$ and $\alpha_{R_{\text{eco}}}$ are subsequently passed to the carbon module in the land surface model in order to scale GPP and $R_{\text{eco}}$.

## 3 CO$_2$ forecast simulations

Several simulations have been performed in order to test the impact of BFAS on the atmospheric CO$_2$ forecasts (see Table 2). All the simulations use the CAMS CO$_2$ forecasting system (Agustí-Panareda et al., 2014) based on the IFS model (www.ecmwf.int/en/forecasts/documentation-and-support). They all share the same transport. The only difference between them is the CO$_2$ surface fluxes they use as described in Table 2. The impact of BFAS is assessed by comparing the simulations using modelled NEE fluxes without BFAS (CTRL) and with BFAS (BFAS). The BFAS simulation is also compared with the simulations using optimised fluxes (OPT) and a climatology of optimised fluxes (OPT-CLIM). Both OPT and OPT-CLIM simulations constitute a benchmark because they are driven by the reference fluxes used in BFAS. From these experiments we expect to see the forecast from BFAS to be closer to the benchmark forecasts (in particular CLIM-OPT) than to the CTRL forecast.

The forecasts are performed using the cyclic configuration described by Agustí-Panareda et al. (2014) with a spectral resolution of TL255, equivalent to around 80 km

Discussion Paper | Discussion Paper | Discussion Paper | Discussion Paper | Discussion Paper

in the horizontal, and 60 vertical levels. They are initialised daily at 00:00 UTC with ECMWF operational analysis, while the atmospheric $CO_2$ is cycled from one forecast to the next, as in a free run. The simulations span the period from 1 January to 31 December 2010. This period has been selected because of the large variety of observations available to evaluate the BFAS performance on the atmospheric $CO_2$ forecasts. The $CO_2$ initial conditions on 1 January 2010 are from the atmospheric $CO_2$ analysis using GOSAT $CO_2$ retrievals (Heymann et al., 2015).

## 4 Monitoring the flux adjustment

The flux adjustment is monitored by plotting time series of the flux scaling factors for each vegetation type and region. For example, Fig. 5 shows the GPP and $R_{eco}$ scaling factors for the crop vegetation type which is present in all regions. The values range from 0.5 to 6. These coefficients are computed daily before the beginning of each forecast and they are kept constant throughout the forecast. Generally, there is a slow variation of the coefficients from one day to the next. This is expected since the coefficients are obtained from large-scale budgets computed over a 10-day period. The map of the GPP and $R_{eco}$ scaling factors applied to adjust the modelled biogenic fluxes on 15 March 2010 is shown in Fig. 6. These maps can be very useful to monitor the flux adjustment because they can provide alerts on the regions with largest biases to model developers.

The effect of the flux adjustment on the NEE budget is shown in Fig. 7. The adjusted biogenic fluxes should always lead to an NEE budget close to the budget of the optimised NEE climatology. However, the fit will also depend on the degree of inter-annual variability of the model determined by parameter $\gamma$ in Eq. (3). Figure 8 displays the monitoring of $\gamma$ given by the standardised NEE anomaly of the model. Positive values mean the $CO_2$ source is larger than normal and/or the $CO_2$ sink is lower than normal with respect to the 10-year mean budget of the model, covering the same period as the reference climatology. Conversely, negative values correspond to a smaller than normal

Discussion Paper | Discussion Paper | Discussion Paper | Discussion Paper | Discussion Paper |

**ACPD**

doi:10.5194/acp-2015-987

**Biogenic flux adjustment scheme for $CO_2$ analysis and forecasting system**

A. Agustí-Panareda et al.

source and/or larger than normal sink. When $\gamma$ is larger than 1, the model anomaly is larger than $1\sigma$. This indicates the possible occurrence of an extreme event. Prolonged extreme events – such as droughts – would have an effect on the NEE budget and the computation of the biogenic flux adjustment.

## 5  Impact of the flux adjustment

The impact of BFAS is shown by comparing the atmospheric $CO_2$ from the BFAS forecast to the CTRL forecast, and to the benchmark forecasts with optimised fluxes (OPT and CLIM-OPT) at several observing sites. Four sites from the NOAA/ESRL atmospheric baseline observatories (www.esrl.noaa.gov/gmd/obop, Thoning et al., 2012) are used to evaluate the reduction of the large-scale biases in the well-mixed background air. In addition, four Total Carbon Column Observing Network stations (GGG2014 TCCON data, Wunch et al., 2011, see Table 3 and www.tccon.caltech.edu) are also used to assess the impact on the atmospheric $CO_2$ column-average dry molar fraction. Finally, three continental sites from the NOAA/ESRL tall tower network (www.esrl.noaa.gov/gmd/ccgg/towers, Andrews et al., 2014) are used to investigate the impact of BFAS on the synoptic skill of the forecasts. The results are grouped into the impacts on bias reduction and synoptic skill in the following two sections.

### 5.1  Biases in atmospheric $CO_2$

Figure 9 demonstrates that BFAS is very effective at reducing the atmospheric $CO_2$ biases in the background air at all the NOAA/ESRL continuous baseline stations. The biases in the CTRL forecast range from −1.9 to −4.5 ppm; whereas, the BFAS forecast has biases of −0.5 ppm or less over the whole year. These values are close to the annual biases of the OPT and OPT-CLIM experiments ranging between −0.4 and 0.5 ppm. The monthly biases in BFAS can be larger than its annual biases. For example, there is a bias of up to −1 ppm from March to September in the southern hemi-

Discussion Paper | Discussion Paper | Discussion Paper | Discussion Paper | Discussion Paper |

**ACPD**

doi:10.5194/acp-2015-987

**Biogenic flux adjustment scheme for CO2 analysis and forecasting system**

A. Agustí-Panareda et al.

sphere (Fig. 9c, d). The bias starts to grow at the end of the growing season during summer time. This is also the case for the high latitude station at Barrow, where there is a negative bias of a few ppm from the last week of July to the end of September as shown in Fig. 9a. In summary, BFAS is not able to completely remove the negative model bias at the end of the growing season. In the northern hemisphere at the end of winter and throughout spring (from March to May) there is a positive model bias, i.e. the atmospheric $CO_2$ is overestimated in the model. Although the OPT and OPT-CLIM simulations also have a slight positive bias in winter, this positive bias is enhanced in the BFAS simulation.

At the TCCON sites (Fig. 10), the atmospheric $CO_2$ column-average dry molar fraction also shows the same large bias reduction in BFAS with respect to CTRL. The magnitude of the BFAS annual biases in the atmospheric column is generally less than 1 ppm, slightly higher than the OPT and OPT-CLIM biases (less than 0.5 ppm), but much lower than the CTRL biases (from 1.5 to 3.3 ppm). The results at the TCCON sites are consistent with those from the NOAA/ESRL baseline sites. Namely, in the northern hemisphere there is a growing overestimation of the atmospheric $CO_2$ at the end of winter (around March). While at the end of the growing season in both northern and southern hemispheres (August and March respectively) there is a growing negative bias, i.e. an overestimation of the sink. One hypothesis that could explain why BFAS is not able to achieve as small a bias as the forecast with optimised fluxes lies in the fact that the optimised NEE used as a reference in BFAS is computed as a residual after removing the effect of fires and anthropogenic fluxes. Inconsistencies in the fire and anthropogenic emissions used by the optimised fluxes and the model will lead to errors in the optimised residual NEE. These inconsistencies are mainly associated with the use of different resolutions. Further investigation is required to address this issue.

## 5.2 Synoptic variability of atmospheric $CO_2$

The $CO_2$ forecast has been shown to have high skill in simulating the synoptic variability of atmospheric $CO_2$ (see Agustí-Panareda et al., 2014), except during the spring

Discussion Paper | Discussion Paper | Discussion Paper | Discussion Paper | Discussion Paper |

**ACPD**

doi:10.5194/acp-2015-987

**Biogenic flux adjustment scheme for CO$_2$ analysis and forecasting system**

A. Agustí-Panareda et al.

months, coinciding with an early start of the $CO_2$ drawdown period in the model. For this reason, we have examined the impact of BFAS on the synoptic variability of daily mean atmospheric $CO_2$ at three continental NOAA/ESRL tower sites in March. Over this period, the day-to-day variability of atmospheric $CO_2$ at those sites is associated

with the advection of atmospheric $CO_2$ by baroclinic synoptic weather systems as they impinge on the large-scale continental gradient of atmospheric $CO_2$. Table 4 clearly demonstrates that with BFAS the synoptic forecast skill is greatly improved at all sites, with correlation coefficients between simulated and observed atmospheric $CO_2$ exceeding 0.8. The improvement is particularly striking at Park Falls (Wisconsin, USA)

and West Branch (Iowa, USA) at the centre of North America, where the correlation coefficients in CTRL are very low (i.e. below 0.5). The OPT and OPT-CLIM forecasts have generally high correlation coefficients, comparable to BFAS. Only at the level closest to the surface, the values are slightly lower than BFAS. This can be explained by the fact that the MACC-13R1 optimised fluxes do not comprise synoptic variability. Thus,

when the synoptic variability of the fluxes contributes to the atmospheric $CO_2$ variability, the correlation coefficients are smaller.

The positive impact of BFAS on the $CO_2$ synoptic variability is illustrated in Fig. 11. The large synoptic variability is characterised by the advection of $CO_2$-rich anomalies (with up to 10 ppm amplitude) as shown by the $CO_2$ peaks on 10–12 March at Park

Falls, and 8–9, 12–13 and 16–17 March at West Branch. These $CO_2$ anomalies originate from the advection across the large-scale continental gradients of atmospheric $CO_2$ which ultimately reflect the large-scale distribution of $CO_2$ surface fluxes (Keppel-Aleks et al., 2012). In the case study here, the $CO_2$-rich air is located to the south of the observing stations, as shown by the distribution of the monthly mean atmospheric

$CO_2$ depicting the large-scale gradients across the continent at the level corresponding to the height of the tall towers (Figs. 12a and 12b). In the CTRL forecast, there is no monthly mean gradient south of the stations (Fig. 12c). This explains why without BFAS the synoptic variability is very small and largely underestimated throughout March. While in BFAS the gradient south of the observing stations is very pronounced

Discussion Paper | Discussion Paper | Discussion Paper | Discussion Paper | Discussion Paper |

**ACPD**

doi:10.5194/acp-2015-987

**Biogenic flux adjustment scheme for $CO_2$ analysis and forecasting system**

A. Agustí-Panareda et al.

**ACPD**

doi:10.5194/acp-2015-987

**Biogenic flux adjustment scheme for CO$_2$ analysis and forecasting system**

A. Agustí-Panareda et al.

(Fig. 12d), following a similar pattern to OPT and OPT-CLIM. There are still some differences between the three simulations. OPT-CLIM results in stronger gradients than OPT and BFAS enhances the gradient even further, leading to a slight over-estimation of the synoptic variability. These differences in the patterns of the atmospheric CO$_2$ are directly linked to the differences in the CO$_2$ surface fluxes (Fig. 13). As expected, the flux adjustment from BFAS results in a flux pattern similar to OPT-CLIM and OPT, with a stronger source to the south of the observing stations. Whereas in CTRL there is a large sink area south of the observing stations, in the region of the Gulf of Mexico, consistent with the CTESSEL early growing season (Balzarolo et al., 2014).

## 6 Discussion

All the results from the BFAS experiments indicate that BFAS is highly beneficial to the CAMS CO$_2$ forecasting system, both in terms of reducing the atmospheric CO$_2$ biases and improving the synoptic skill of the model. As shown in Sect. 2, the scheme is simple and it is easy to implement and run. Because BFAS essentially works as a layer on top of the model, it can adapt to model changes with great flexibility. For all these reasons, BFAS is now part of the operational global CAMS analysis and forecasting system.

Notwithstanding all the advantages of BFAS listed above, there are also caveats that need to be considered, further tested and addressed. A discussion of the current limitations of BFAS is provided in this section, together with the potential use of BFAS for model development and data assimilation purposes.

### 6.1 Current limitations in BFAS

Optimised fluxes have uncertainties of their own and represent the large-scale variability of the CO$_2$ surface fluxes on supra-synoptic time-scales. They only estimate the total flux and the NEE residual approach can transfer biases from other fluxes into the NEE.

The use of a climatology also precludes the correction of the inter-annual variability in the model.

The aggregation criteria of budget errors can be very challenging because the error can originate from different aspects of the model. Clearly, errors in model parameters associated with vegetation type are a good candidate. However, in the future errors in climate forcing, errors in LAI, missing processes and other potential sources of error should also be considered.

The partition of the NEE flux adjustment into the modelled biogenic fluxes (GPP and $R_{eco}$) is currently ad-hoc, leading to the transfer of errors from GPP to $R_{eco}$ and vice-versa. This problem could be addressed by using other independent datasets of GPP and $R_{eco}$ (e.g. Jung et al., 2011) that contain additional information on how to partition the NEE adjustment.

## 6.2  BFAS for model development

BFAS can run in both online and offline modes. Thus, it can provide a tool to diagnose regions that contribute to the errors in the global budget resulting in large-scale errors of atmospheric $CO_2$. The maps of biogenic flux scaling factors can be used to compute maps of flux adjustment (e.g. adjusted NEE – original NEE) which can then be used to diagnose model errors. The synthesis of the mean adjustments into monthly model biases for different vegetation types can then guide the effort to develop the carbon model further. For example, in regions where the bias is consistent between different months, the corrected NEE could be used to re-tune model parameters such as the reference ecosystem respiration or the mesophyll conductance, previously optimised by Boussetta et al. (2013) using a subset of FLUXNET data. Specific vegetation types can be identified where model improvements could be achieved by using information from BFAS. For instance, crops have the same large $R_{eco}$ scaling ($> 1.5$) over all the northern hemisphere regions during winter months when the ecosystem respiration is the dominant component of NEE. This underestimation in the ecosystem respiration can be addressed by modifying the value of the reference respiration parameter used

Discussion Paper | Discussion Paper | Discussion Paper | Discussion Paper | Discussion Paper |

**ACPD**

doi:10.5194/acp-2015-987

**Biogenic flux adjustment scheme for CO2 analysis and forecasting system**

A. Agustí-Panareda et al.

**ACPD**

doi:10.5194/acp-2015-987

**Biogenic flux adjustment scheme for CO$_2$ analysis and forecasting system**

A. Agustí-Panareda et al.

for crops. In this case, the same procedure used by Boussetta et al. (2013) could be applied to optimise the specific model parameter using the BFAS adjusted fluxes as pseudo-observations together with the FLUXNET data.

Further information on error sources in fluxes can be obtained by comparing the corrected fluxes with the eddy covariance observations available in near-real time from the Integrated Observation System (ICOS) Ecosystem Thematic Centre (ETC, http://www.europe-fluxdata.eu). For example, preliminary comparisons have shown that there are large differences in the model-observation fit between needle leaf evergreen (pine) trees in the boreal and Mediterranean regions. This is consistent with results from Balzarolo et al. (2014), and it highlights the need for a new sub-classification of the evergreen needle leaf forests in regions with Mediterranean climate.

## 6.3 BFAS in the data assimilation framework

Currently, BFAS is only designed to be used as a bias correction computed before each forecast by using a reference data set based on optimised fluxes. In the future, BFAS could be adapted to work within a data assimilation (DA) framework in the IFS. To start with, the use of uncertainties associated with both the reference data set and the model would allow a more optimal estimation of the flux adjustment. These uncertainties can be obtained from the flux inversion systems for the optimised fluxes and from the ECMWF ENS forecasts for the model fluxes.

Including BFAS in the IFS DA framework needs further exploration. The IFS uses a short time window (currently 12 h) to assimilate meteorological observations from very dense observing networks. With the short time window it is not possible to properly constrain the slowly varying global mass of the long-lived greenhouse gases due to the sparseness of their observing system. For instance, the current GOSAT and OCO-2 CO$_2$ observations do not cover high latitudes in winter. However, if we combined the assimilation of optimised fluxes (which already contain the global mass constraint) with observations linked to local fluxes (e.g. solar-induced chlorophyll fluorescence products from satellites, NEE eddy covariance observations and in situ atmospheric CO$_2$

Discussion Paper | Discussion Paper | Discussion Paper | Discussion Paper

observations) it might be possible to obtain an optimal estimate of more local scaling factors, while still respecting the global mass constraint. The possibility of optimising the scaling factors in the DA system within the weak constraint framework (Trémolet, 2006, 2007) also needs to be explored in the future.

## 7 Summary

A new biogenic flux adjustment scheme (BFAS) has been developed at ECMWF to reduce large-scale biases of the ecosystem fluxes modelled by the CTESSEL carbon model. This is achieved by a simple scaling of the 10-day NEE budgets for different vegetation types and regions using a climatology of the MACC optimised fluxes (Chevallier et al., 2010) as a reference, adjusted to preserve the model inter-annual variability.

This paper shows that BFAS has a positive impact on the atmospheric $CO_2$ forecast by greatly reducing the atmospheric $CO_2$ biases in background air and improving the synoptic variability in continental regions affected by ecosystem fluxes. The improvement in the synoptic skill of the forecast is associated with underlying changes in the large-scale gradient of the NEE fluxes where optimised fluxes provide information. Because of its simplicity, adaptability to model changes and beneficial impact, BFAS has been recently implemented in the CAMS operational $CO_2$ forecast and analysis system. As a diagnostic tool, BFAS has also potential for model development. The use of BFAS in the data assimilation framework will be explored in the future.

*Acknowledgements.* This study has been funded by the European Commission under the Monitoring of Atmospheric Composition and Climate (MACC) project and the Copernicus Atmosphere Monitoring Service (CAMS).

TCCON data were obtained from the TCCON Data Archive, hosted by the Carbon Dioxide Information Analysis Center (CDIAC) – tccon.onrl.gov. The authors would like to acknowledge the PIs of the different TCCON stations used in this study: Rigel Kivi (Sodankylä, Finland), Nicholas Deutscher (Bialystok, Poland), Paul Wennberg (Lamont, USA) and David Griffith (Wollongong, Australia).

Discussion Paper | Discussion Paper | Discussion Paper | Discussion Paper

**[ACPD](doi:10.5194/acp-2015-987)**

doi:10.5194/acp-2015-987

**Biogenic flux adjustment scheme for CO$_2$ analysis and forecasting system**

A. Agustí-Panareda et al.

The NOAA/ESRL Global Monitoring Division data from the baseline observatories at Barrow (Alaska, USA), Mauna Loa (Hawaii, USA), American Samoa (USA), South Pole (Antarctica), as well as the tall towers at Argyle (Maine, USA), Park Falls (Wisconsin, USA) and West Branch (Iowa, USA) were obtained from ftp://aftp.cmdl.noaa.gov/data/greenhouse_gases/co2.

The authors are very grateful to Nils Wedi for his support in processing the Olson ecosystem classification maps and to Frederic Vitard for providing support and advice on the use of the ENS hindcasts.

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

**Table 1.** Percentage of land grid points at model resolution TL255 ($\sim 80$ km) for each dominant vegetation type, i.e. more than half of the grid point is covered by that vegetation type. A land grid point is defined by a land sea mask value greater than 0.5.

| Vegetation Code | Vegetation type | Percentage of land points |
|---|---|---|
| 1 | Crops, Mixed Farming | 9.9 |
| 2 | Short Grass | 7.6 |
| 7 | Tall Grass | 6.3 |
| 9 | Tundra | 6.3 |
| 10 | Irrigated Crops | 2.2 |
| 11 | Semidesert | 13.5 |
| 13 | Bogs and Marshes | 0.8 |
| 16 | Evergreen Shrubs | 0.5 |
| 17 | Deciduous Shrubs | 2.4 |
| 3 | Evergreen Needle leaf Trees | 5.7 |
| 4 | Deciduous Needle leaf Trees | 2.4 |
| 5 | Deciduous Broadleaf Trees | 4.0 |
| 6 | Evergreen Broadleaf Trees | 12.1 |
| 18 | Mixed Forest/woodland | 3.3 |
| 19 | Interrupted Forest | 9.5 |
| 21 | Tropical Savanna (new type) | 4.8 |
| – | Remaining land points without vegetation | 8.7 |

Discussion Paper | Discussion Paper | Discussion Paper | Discussion Paper

**ACPD**

doi:10.5194/acp-2015-987

**Biogenic flux adjustment scheme for CO₂ analysis and forecasting system**

A. Agustí-Panareda et al.

**ACPD**

doi:10.5194/acp-2015-987

**Biogenic flux adjustment scheme for CO$_2$ analysis and forecasting system**

A. Agustí-Panareda et al.

**Table 2.** List of simulations with the same transport and different CO$_2$ surface fluxes.

| Experiment name | CO$_2$ surface fluxes |
|---|---|
| CTRL | Biogenic fluxes from CTESSEL (Boussetta et al., 2013), biomass burning fluxes from GFAS (Kaiser et al., 2012), ocean fluxes from Takahashi et al. (2009), and EDGAR v4.2 anthropogenic fluxes (Janssens-Maenhout et al., 2012) |
| OPT | MACC-13R1 optimised fluxes (Chevallier et al., 2010) for 2010 |
| CLIM-OPT | MACC-13R1 optimised flux climatology (2004–2013) as the reference in BFAS |
| BFAS | Same fluxes as CTRL including BFAS |

**ACPD**

doi:10.5194/acp-2015-987

**Biogenic flux adjustment scheme for CO$_2$ analysis and forecasting system**

A. Agustí-Panareda et al.

**Table 3.** List of TCCON stations used in Fig. 10 ordered by latitude from North to South.

| Site | Latitude [degrees] | Longitude [degrees] | Altitude [m a.s.l] | Reference |
|------|------|------|------|------|
| Sodankylä | 67.37 | 26.63 | 190.0 | Kivi et al. (2014) |
| Białystok | 53.23 | 23.02 | 160.0 | Deutscher et al. (2014) |
| Lamont | 36.60 | −97.49 | 320.0 | Wennberg et al. (2014) |
| Wollongong | −34.41 | 150.88 | 30.0 | Griffith et al. (2014) |

# ACPD

doi:10.5194/acp-2015-987

**Biogenic flux adjustment scheme for CO$_2$ analysis and forecasting system**

A. Agustí-Panareda et al.

**Table 4.** Correlation coefficient of different forecast (FC) experiments (see Table 2) with observations at three NOAA/ESRL tall towers for daily mean dry molar fraction of atmospheric CO$_2$ in March 2010. The dash symbol means the correlation is not significant.

| NOAA/ESRL Tower site (ID) | Latitude, Longitude, Altitude | Sampling level [m] | BFAS FC | CTRL FC | OPT FC | OPT-CLIM FC |
|---|---|---|---|---|---|---|
| Park Falls, Wisconsin (LEF) | 45.95° N, 90.27° W, 472 m | 30 | 0.843 | 0.338 | 0.794 | 0.797 |
| | | 122 | 0.931 | 0.508 | 0.893 | 0.883 |
| | | 396 | 0.919 | – | 0.875 | 0.881 |
| West Branch, Iowa (WBI) | 41.72° N, 91.35° W, 242 m | 31 | 0.748 | 0.496 | 0.590 | 0.590 |
| | | 99 | 0.833 | 0.436 | 0.767 | 0.720 |
| | | 379 | 0.851 | 0.356 | 0.887 | 0.876 |
| Argyle, Maine (AMT) | 45.03° N, 68.68° W, 50 m | 12 | 0.857 | 0.839 | 0.808 | 0.893 |
| | | 30 | 0.875 | 0.835 | 0.816 | 0.938 |
| | | 107 | 0.861 | 0.668 | 0.816 | 0.927 |

Discussion Paper | Discussion Paper | Discussion Paper | Discussion Paper |

**ACPD**

doi:10.5194/acp-2015-987

**Biogenic flux adjustment scheme for CO$_2$ analysis and forecasting system**

A. Agustí-Panareda et al.

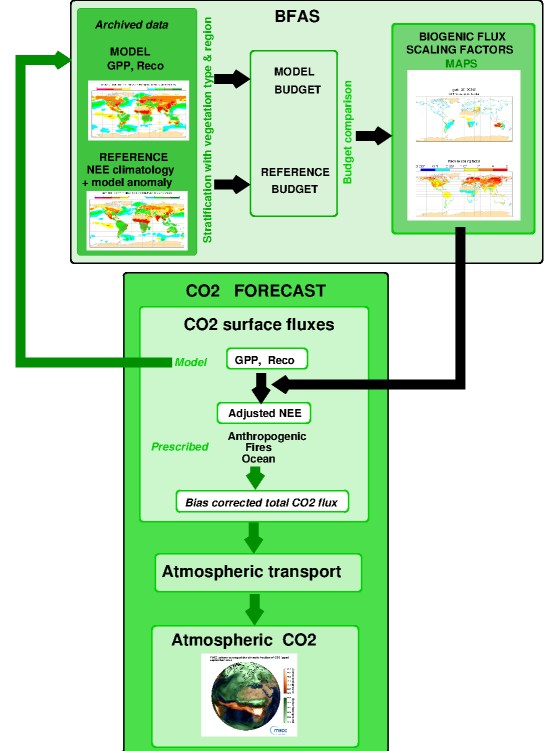

**Figure 1.** Schematic showing how BFAS fits in the atmospheric CO$_2$ forecasting system. BFAS is called before each forecast to compute the scaling factors for the model NEE (i.e. GPP + $R_{eco}$) based on the past archived forecasts. The maps of the scaling factors are then passed to the model which applies the adjustment to the output biogenic CO$_2$ fluxes from the land surface model. After combining the adjusted NEE fields with the other prescribed CO$_2$ fluxes, the resulting bias corrected fluxes are passed to the transport model to produce the atmospheric CO$_2$ forecast.

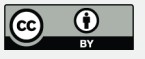

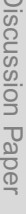

**ACPD**

doi:10.5194/acp-2015-987

**Biogenic flux adjustment scheme for CO$_2$ analysis and forecasting system**

A. Agustí-Panareda et al.

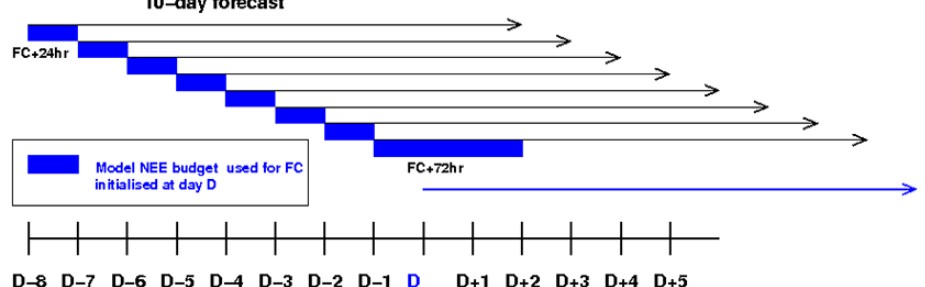

**Figure 2.** Schematic to illustrate how the 10-day NEE budget from the model is computed in BFAS for the forecast at day $D$ by retrieving the past forecasts of accumulated NEE. Note that the retrieved NEE (computed by adding GPP and $R_{eco}$) has not been corrected by BFAS. The computation uses a set of 7 previous 1-day forecasts (initialised at $D-8$, $D-7$, $D-6$,... until $D-2$) together with the latest 3-day forecast from the previous day (i.e. $D-1$) as shown by the blue boxes.

**[ACPD]**

doi:10.5194/acp-2015-987

**Biogenic flux adjustment scheme for CO$_2$ analysis and forecasting system**

A. Agustí-Panareda et al.

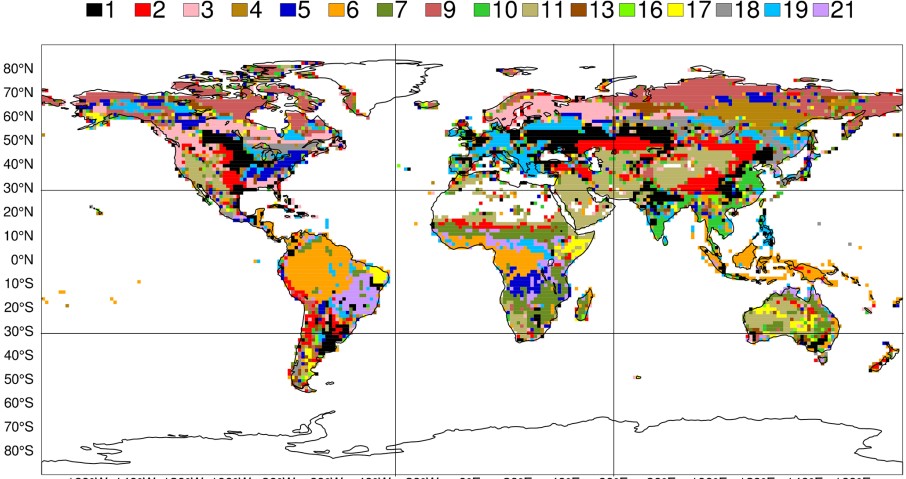

**Figure 3.** Dominant vegetation types based on the BATS classification used in the IFS and extended to include the tropical savanna subtype (in purple, as defined by the Olson (1994a) classification) within the interrupted forest type (in light blue). The vegetation type codes are described in Table 1. The nine regions used in the computation of the NEE budget are delimited by the black lines.

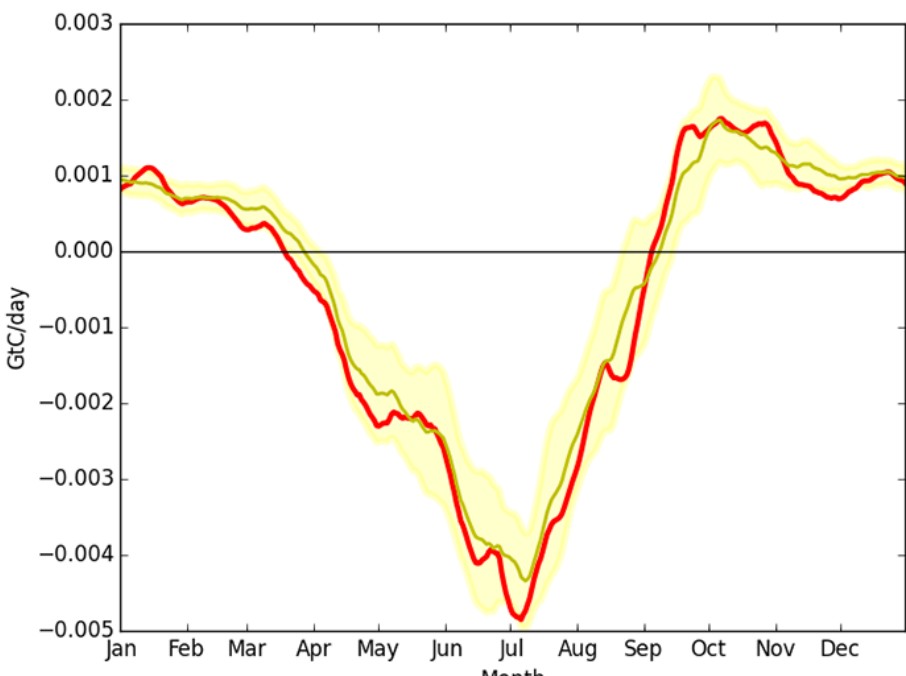

**Figure 4.** Time series of 10-day mean NEE budget [GtC/day] associated with the crop vegetation type in North America from the MACC-13R1 optimised flux data set in 2010 (red line) compared to its climatology (2004–2013) (yellow line). The yellow shading represents the standard deviation of the optimised flux budget (for the same period) used to compute the inter-annual variability adjustment applied to the reference climatology. Positive/negative values correspond to a source/sink of $CO_2$.

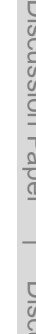

# ACPD

doi:10.5194/acp-2015-987

**Biogenic flux adjustment scheme for CO$_2$ analysis and forecasting system**

A. Agustí-Panareda et al.

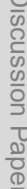

**ACPD**

doi:10.5194/acp-2015-987

**Biogenic flux adjustment scheme for CO$_2$ analysis and forecasting system**

A. Agustí-Panareda et al.

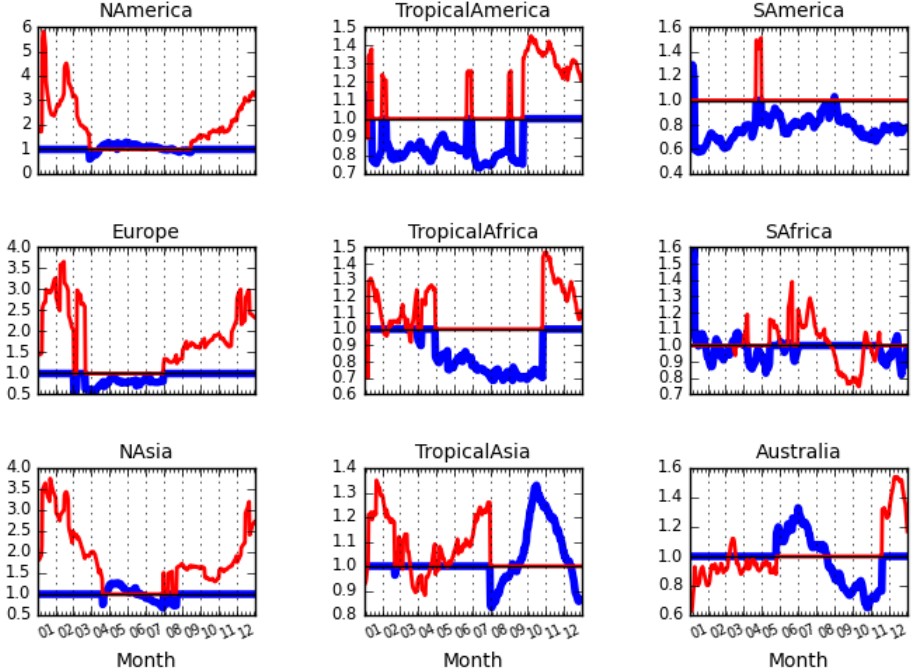

**Figure 5.** Time series of GPP and $R_{\mathrm{eco}}$ flux scaling factors in blue and red lines respectively for the crop vegetation type in 2010 in the different regions (see map in Fig. 3 depicting the extent of the crops within each region).

**ACPD**

doi:10.5194/acp-2015-987

**Biogenic flux
adjustment scheme
for CO$_2$ analysis and
forecasting system**

A. Agustí-Panareda et al.

(a) GPP scaling factor

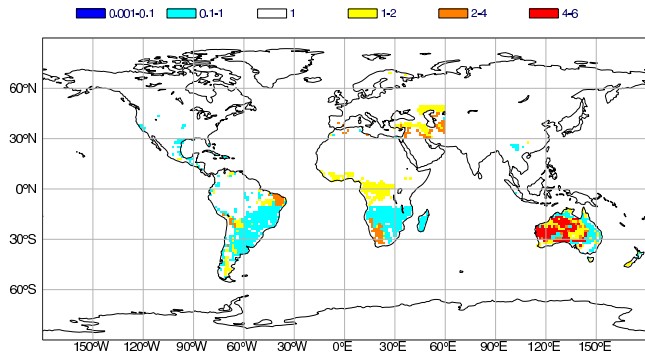

(b) Reco scaling factor

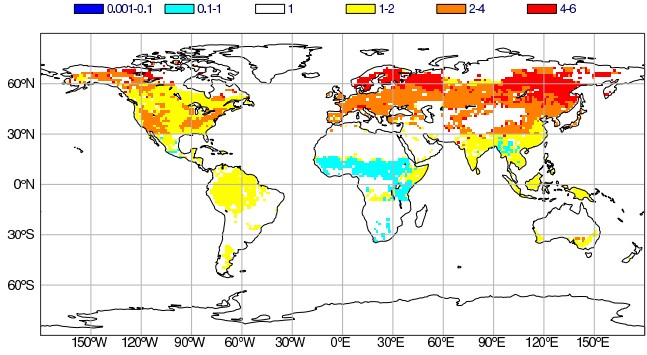

**Figure 6.** Map of scaling factors for **(a)** GPP and **(b)** $R_{\text{eco}}$ on 15 March 2010.

**ACPD**

doi:10.5194/acp-2015-987

**Biogenic flux adjustment scheme for CO$_2$ analysis and forecasting system**

A. Agustí-Panareda et al.

Discussion Paper | Discussion Paper | Discussion Paper | Discussion Paper

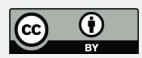

**Figure 7.** Time series of GPP (in blue), $R_{eco}$ (in red) and NEE (in green) daily budget [GtC/day] before and after the flux adjustment (see dashed lines and solid lines respectively) for crops in 2010 in the different regions. The reference budget provided by the climatology of MACC-13R1 optimised fluxes (2004–2013) and the MACC-13R1 optimised fluxes for 2010 are depicted by the black and magenta lines respectively. Positive/negative values correspond to a source/sink of CO$_2$.

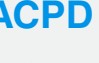

**ACPD**

doi:10.5194/acp-2015-987

**Biogenic flux adjustment scheme for CO$_2$ analysis and forecasting system**

A. Agustí-Panareda et al.

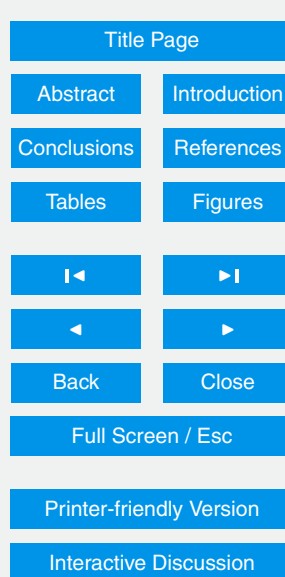

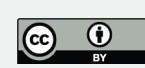

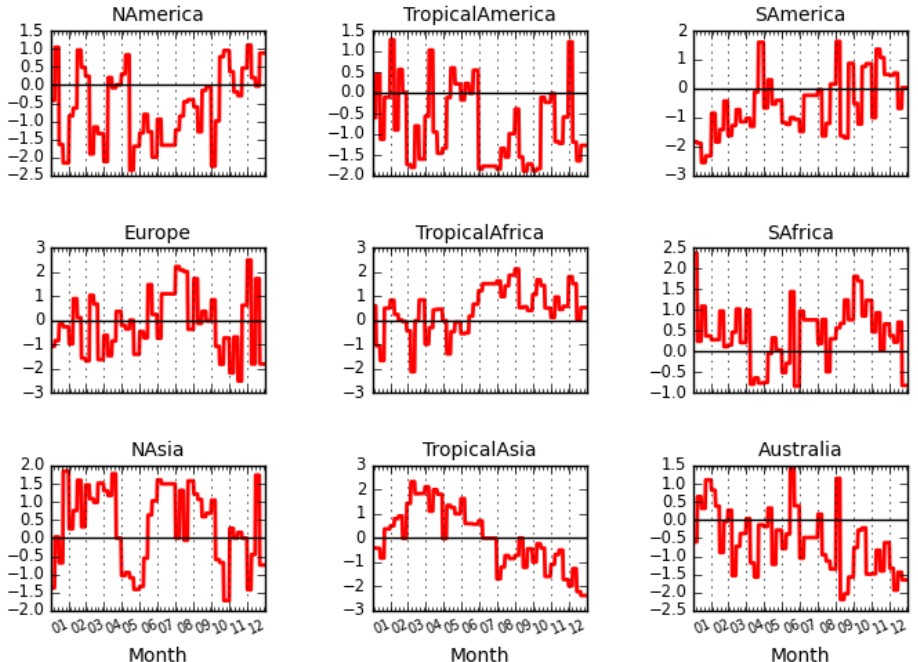

**Figure 8.** Time series of the standardised anomaly of the modelled NEE budget ($\gamma$ in Eq. 3) for crops in 2010 in the different regions. Positive values indicate larger/smaller CO$_2$ sources/sinks than normal based on the mean climatological budget; whereas negative values correspond to smaller/larger CO$_2$ sources/sinks than normal.

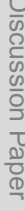

# ACPD

doi:10.5194/acp-2015-987

**Biogenic flux adjustment scheme for CO$_2$ analysis and forecasting system**

A. Agustí-Panareda et al.

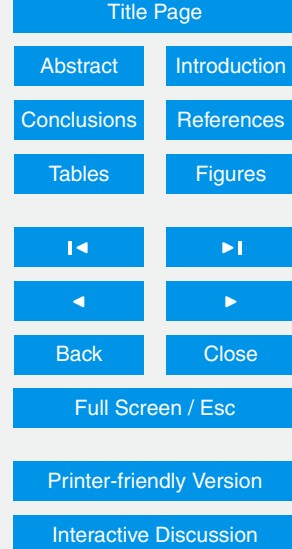

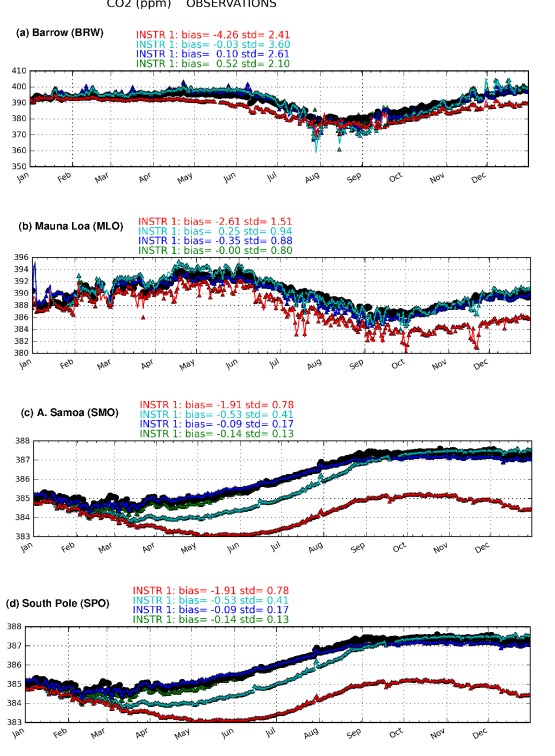

**Figure 9.** Daily mean atmospheric CO$_2$ dry molar fraction [ppm] from NOAA/ESRL continuous baseline stations (black circles) at **(a)** Barrow, Alaska, USA (71.32° N, 156.61° W), **(b)** Mauna Loa, Hawaii, US (19.54° N, 155.58° W), **(c)** Tutuila, American Samoa, USA (14.25° S, 170.56° W), **(d)** South Pole, Antarctica (89.98° S, 24.8° W) and the different forecast experiments: BFAS (cyan), CTRL (red), OPT (green) and OPT-CLIM (blue). See Table 2 for a description of the different experiments. The mean (bias) and standard deviation (SD) of the model errors are shown at the top of each panel.

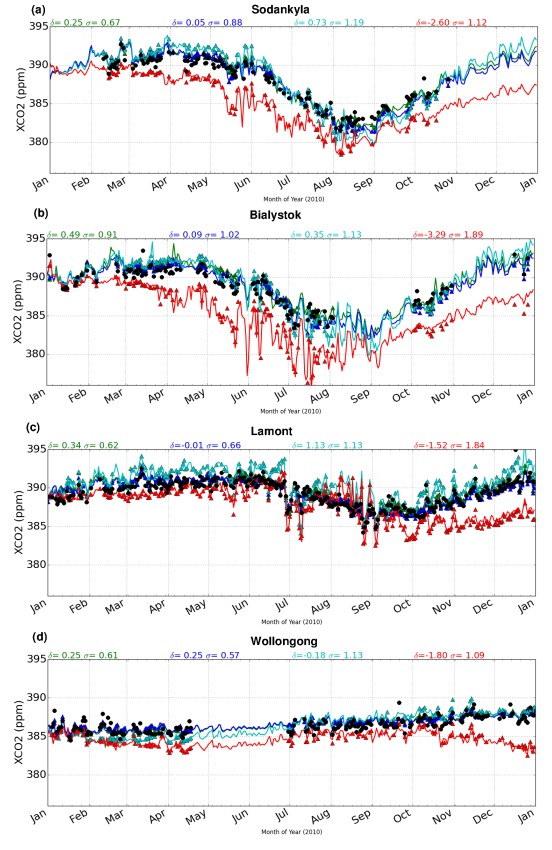

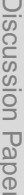

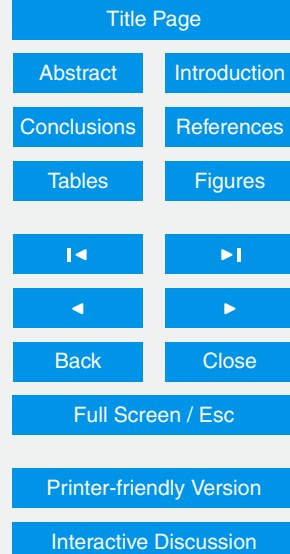

**Figure 10.** Daily mean atmospheric $CO_2$ column-average dry molar fraction [ppm] observed at four TCCON stations (see Table 3) as shown by the black circles, and simulated by the different forecast experiments: BFAS (cyan), CTRL (red), OPT (green) and OPT-CLIM (blue). See Table 2 for a description of the different experiments. The mean ($\delta$) and standard deviation ($\sigma$) of the model errors are shown at the top of each panel.

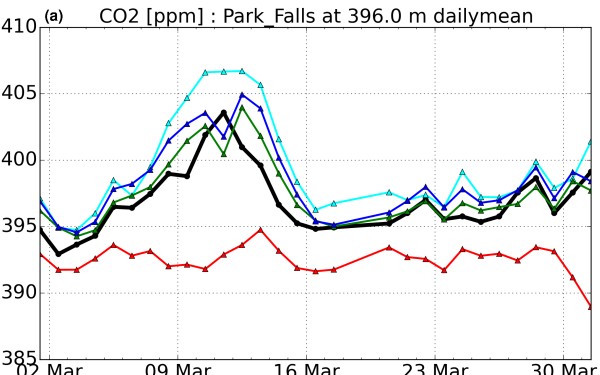

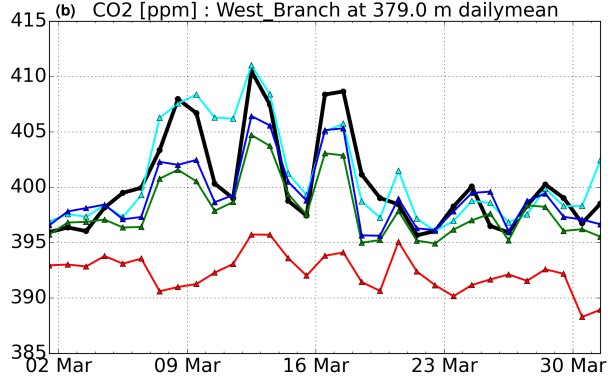

**Figure 11.** Daily mean atmospheric $CO_2$ dry molar fraction [ppm] in March 2010 from NOAA/ESRL tall towers (black circles) at **(a)** Park Falls (Wisconsin, USA, 45.95° N, 90.27° W) and **(b)** West Branch (Iowa, USA, 41.72° N, 91.35° W) and the different forecast experiments: BFAS (cyan), CTRL (red), OPT (green) and OPT-CLIM (blue) (see Table 2 for a description of the different experiments).

**ACPD**

doi:10.5194/acp-2015-987

**Biogenic flux adjustment scheme for CO$_2$ analysis and forecasting system**

A. Agustí-Panareda et al.



**ACPD**

doi:10.5194/acp-2015-987

**Biogenic flux adjustment scheme for CO$_2$ analysis and forecasting system**

A. Agustí-Panareda et al.

**(a) OPT−CLIM FC**

**(b) OPT FC**

**(c) CTRL FC**

**(d) BFAS FC**

**Figure 12.** Monthly mean atmospheric CO$_2$ dry molar fraction [ppm] at the model level approximately corresponding to the highest sampling height of the Park Falls and West Branch NOAA/ESRL tall towers (see black triangles) in March 2010 from **(a)** OPT-CLIM, **(b)** OPT, **(c)** CTRL and **(d)** BFAS experiments (see Table 2 for a description of the different experiments).

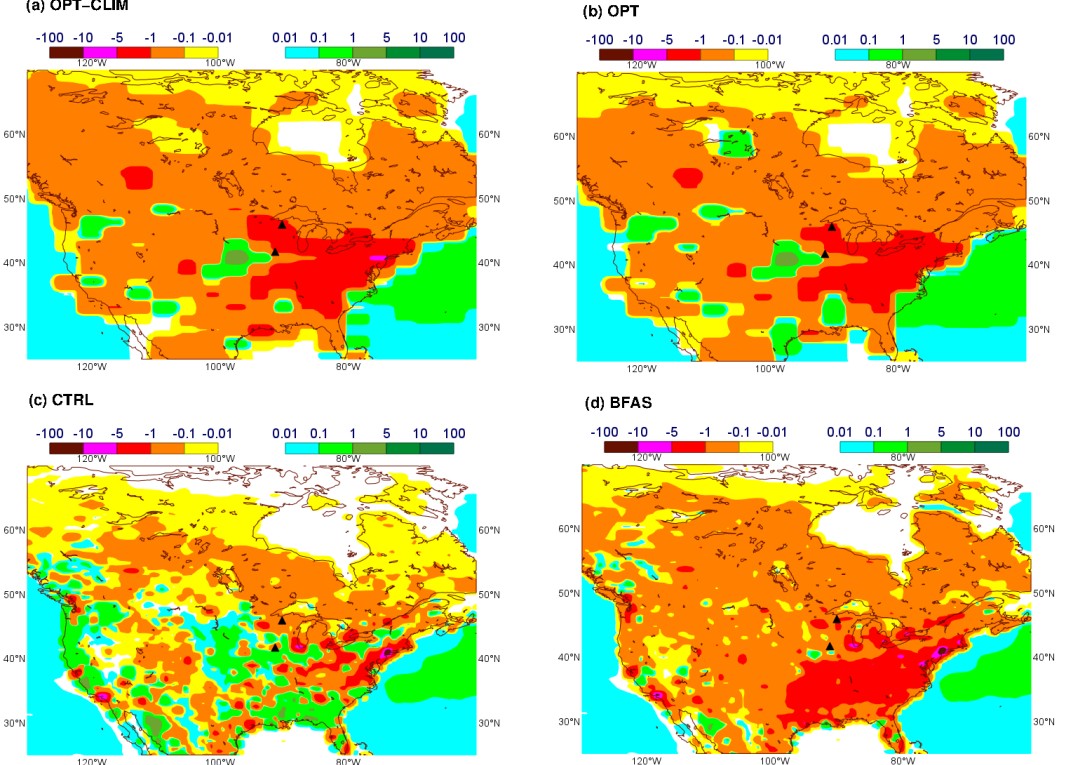

**Figure 13.** Monthly mean total $CO_2$ flux [µmol m$^{-2}$ s$^{-1}$] in March 2010 from **(a)** OPT-CLIM, **(b)** OPT, **(c)** CTRL and **(d)** BFAS experiments (see Table 2 for a description of the different experiments). The black triangles depict the location of the NOAA/ESRL tall towers plotted in Fig. 11.

Discussion Paper | Discussion Paper | Discussion Paper | Discussion Paper | Discussion Paper |

**ACPD**

doi:10.5194/acp-2015-987

**Biogenic flux adjustment scheme for CO₂ analysis and forecasting system**

A. Agustí-Panareda et al.