# Peer review of "A biogenic CO2 flux adjustment scheme for the mitigation of large-scale biases in global atmospheric CO2 analyses and forecasts"

_Atmospheric Chemistry and Physics, 2015_

## Referee Comment (RC1) · Anonymous Referee #1 · 10 Feb 2016

This paper presents an improvement of the simulated atmospheric CO2 budget within the IFS/CAMS modeling system. This improvement consists of an adjustment of the underlying biospheric CO2 fluxes (GPP, TER, and NEE). The authors use a scheme that relies on pre-calculated values of these fluxes from a separate inverse model. These fluxes were created as part of a joint project. The authors show the improvement of the new CO2 budget over the old one and try to show that this new system has advantages when forecasting atmospheric CO2 especially at the synoptic time scale. This works through a better representation of large scale zonal gradients of CO2, that come to observing sites strongly when the synoptic weather patterns change. Finally,

the authors discuss future steps for this system. This paper is written well and structured and represents a large piece of development of the CAMS model.

Nevertheless, in my opinion this paper has a number of problems and I believe that it is not currently suitable for publication in ACP. The first is that the paper contains relatively little scientific content, and there is nearly nothing that can be learned from the paper for a big audience. And even for researchers in the field of atmospheric $CO_2$ modeling, these methods are very system specific and not easily used by others even if they needed such flux adjustments. So this paper should probably remain a technical report for the Copernicus project, or perhaps it can be published in Geophysical Model Development journal. The case of why having better synoptic variations in forecast $CO_2$ is important is also not clearly made I think: who or what profits from this improved $CO_2$ forecast?

Another issue with the paper is the choice of the control run. Taking the fluxes from the neutral-biosphere in CTESSEL is clearly wrong, and there could have been many easy ways to improve on those. I think that a better benchmark is the available MACC fluxes, as the authors show that these already do quite a good job in matching observations if simply prescribed to the CAMS model. The authors state that these fluxes do not have synoptic variability, and I am not clear why this is because their resolution is never mentioned in the paper. But if diurnal and synoptic variations are needed, the simple method of Olson and Randerson (2004) can be used to include the effect of temperature and light on monthly mean fluxes to get hourly ones. If the BAFS system was shown to be better than such an offline flux product, it would be much more clear to me that this way of BAFS is the way forward for CAMS.

In this manuscript, it is not clear to me why certain metrics were chosen for evaluation. The authors present mean biases and standard deviations in Figures 9 and 10, correlation coefficients in Table 4, no metric for Figure 11, but there are never root-mean-square differences reported which I think are most useful. I think in figure 11 the MACC fluxes have the lowest RMSD than the BFAS fluxes. And from the captions it

seems that both observations and simulations are done as daily (24-hour?) averages. I think that this daily averaging is needed because the independent adjustment of the GPP and TER scaling factors leads to strong variations in NEE that do not necessarily preserve a good diurnal cycle. But I might be wrong on that, as I could not assess this from the figures shown. 24-hour average observations could have a lot of hour-to-hour variability which should be shows by an error bar. The statistics and figures moreover seem to cover only the month of March and a few selected days in March. It remains unexplained why this choice was made, and what the metrics look like for other months. I would expect for instance in summer to see even larger day-to-day variations in NEE, and then also in atmospheric CO₂.

I would like to know what the added value is of having the gamma-parameter included in BFAS. The description of its calculation and adjustment is quite extensive but I do not really understand what role it plays. Perhaps there could be an experiment where BFAS is used without the adjustment in equation 3. After all, not needing the ensemble of forecasts would make the scheme a bit simpler, and perhaps just as good? I know I am likely to be wrong as the authors have decided to include this procedure in BFAS, but I would like to see the evidence to support that decision.

A further list of minor comments can hopefully help the authors prepare a new version for their manuscript if they want to submit it somewhere else.

- Page 3, line 5: I do not agree that the current monitoring of CO2 relies on satellites and it is even a bit insulting to the real monitoring groups to say it. I suggest to change it because satellites do not yet see reliable CO₂. In fact, the second part of this statement is also not right because the observations you show and that MACC fluxes rely on mostly come from flasks and not from in-situ instruments. - Page 12, line 20: the current adjustment scheme for GPP and TER does not include any covariances between the adjustments, but we know that they often respond in the same direction and that errors are correlated. It would be good to think about an adjustment scheme that uses such information. Showing the posterior diurnal cycle is also needed. - Page

13, line 20: You use now the names OPT-CLIM and later on in the text and tables CLIM-OPT. Is this the same run? It was to me confusing. Also see later remark about Table 2. - Page 14, line 20: A table listing the annual mean fluxes for transcom regions for all simulations would be valuable I think - Page 15, line 25: The SH problems could come from a different north to south transport characteristic of the two atmospheric models used (IFS and LMDZ?). Can this be illustrated with a simple SF6 simulation and compare it to observations? - Acknowledgements: please check the data usage policy of NOAA as I do not believe you can simply take data from their FTP and then publish it with this statement. - Page 30, Table 2: I was confused because it says that CLIM-OPT uses MACC fluxes as reference in BFAS but from the methods I understood that CLIM-OPT or OPT-CLIM used the climatological fluxes from MACC directly as underlying biosphere fluxes? I discovered this only towards the end of reading and it made me think I misunderstood the simulations completely. Even now I doubt it. - Figures 4 and 7: it would be better to use PgC/yr as units and not GtC/day because now they just look very small on the y-axis with many insignificant digits to start. - I believe Figure 12 and 13 are not needed and could be removed

---

## Referee Comment (RC2) · P. Rayner (Referee) · 8 Mar 2016

This paper presents an enhancement to the CO2 assimilation system used within the Copernicus tracer assimilation system at ECMWF. The enhancement is certainly useful and potentially quite important but it comes with its own problems. I believe these need to be discussed in the manuscript and addressed in how the new product is made available.

The enhancement addresses the problem of large-scale biases in the fluxes which underlie the prior concentrations used in the assimilation. These biases are a serious

matter since they mean that the probability densities assumed in the assimilation system (centered on the true value) don't, in fact, hold. So this is a potentially valuable improvement.

The problem arises when we consider what the generated $CO_2$ fields are used for. Although there is probably some benefit for improved retrievals of temperature and moisture by improving the $CO_2$ field the overwhelming use for the assimilated $CO_2$ products is in estimating surface fluxes. the statistical apparatus is identical to the assimilation of the $CO_2$ fields and the same restrictions apply. Among them is a firm prohibition on reusing information and the requirement that observations and prior are independent. Both of these are potentially violated in any downstream use of the BFAS product. Let's deal with these two problems in turn.

The assimilated $CO_2$ field now includes information from a prior informed by a previous flux inversion. This inversion presumably used measurements from the in situ network, aircraft and/or TCCON. We can't tell which without a detailed examination of the papers that underlie that inversion. We need to know because, if we're going to use the BFAS product to drive a future inversion, we need to exclude those measurements. One might argue that the periods don't overlap but the evidence of the paper shows that the model-data mismatch is so strongly correlated from year to year (consistent seasonal errors in the pre-BFAS version) that this doesn't avoid the problem.

The second problem, of the prior estimate for a flux inversion being partially reflected in the data we use is not new with BFAS. It exists in the original Copernicus products too. I'm unsure whether the mixing data and model information in the prior $CO_2$ field makes this problem worse but it seems like it should.

Finally there is the question of the uncertainty of the BFAS $CO_2$ field. There are two countervaling effects in play. First the bias correction of the prior has reduced residuals in the generated $CO_2$ field so that uncertainties (which are the statistics of the difference between estimated and true values) seem to have reduced. On the other hand an

extra process has been added to the assimilation with a new set of parameters to scale prior fluxes. These will have their own uncertainty and should (since the posterior $CO_2$ field is sensitive to its prior) increase posterior uncertainty. Which of these wins out? I am always a little wary of criticizing a paper for things it did not do since no piece of research is complete. However it's an important general rule that products that are to be used as inputs to statistical procedures such as flux inversions need to specify their uncertainty as well as their mean.

I believe this paper is a potentially valuable contribution and look forward to the authors' revision. If the authors accept my first point about the mixing of data into their $CO_2$ field then they also need to find a way of detailing which data was used to generate the flux fields that underlie BFAS.

---

## Author Comment (AC1) · 6 May 2016

We thank the reviewer for his/her comments. We will take them into account in the revised manuscript to improve the motivation and the message of this work. In particular, we will highlight the scientific content of our results. In the reply below we address all the reviewer's concerns in order to clarify any misunderstanding on the importance of this study, and its relevance for the scientific community working on atmospheric composition and the carbon cycle.

**General comments**

*\* In my opinion this paper has a number of problems and I believe that it is not currently suitable for publication in ACP. The first is that the paper contains relatively little scientific content, and there is nearly nothing that can be learned from the paper for a big audience. And even for researchers in the field of atmospheric $CO_2$ modeling, these methods are very system specific and not easily used by others even if they needed such flux adjustments. So this paper should probably remain a technical report for the Copernicus project, or perhaps it can be published in Geophysical Model Development journal. The case of why having better synoptic variations in forecast $CO_2$ is important is also not clearly made I think: who or what profits from this improved $CO_2$ forecast?*

The major aspects raised by the reviewer are addressed separately in detail below:

1. The scientific content of the paper.

   Any atmospheric $CO_2$ forecast system requires a flux adjustment of some sort in order to constrain the budget of sources/sinks at the surface and avoid the growth of biases in the atmospheric background as documented by Agusti-Panareda et al. (2014). The scientific question addressed in this paper is how to use the best information we have in near-real time to adjust the fluxes in a way that reduces the bias of the atmospheric $CO_2$ forecast with the minimum deterioration of the synoptic skill. The simple flux adjustment scheme proposed here is based on a climatology of optimized fluxes and it could be applied easily to other models. In the past other methods have been used by several modelling studies to remove biases attributed to the NEE fluxes. For instance, by globally re-scaling balanced NEE fluxes to match the residual land sink given by a climatology of TRANSCOM optimized fluxes (Nassar et al., 2010; Chen et al.,2013), or by re-scaling locally the NEE at boreal regions in order to get a better fit in the seasonal cycle (e.g.

Messerschmidt et al. 2013, Keppel-Aleks et al. 2012).

This paper addresses the challenge of designing an online bias correction in a forecasting system with the aim to deliver an atmospheric $CO_2$ forecast and analysis that can be useful to the scientific community. The other methods mentioned previously are designed to work as a one-off correction and they offer less flexibility because they are performed offline. Tunning model parameters and/or rescaling fluxes offline are not sufficient to garantee a bias reduction in the system. An online adaptive system is required because errors in the meteorology can evolve as a result of regular operational Numerical Weather Prediction (NWP) model upgrades and these affect the NEE budget in the model.

From the flux adjustment method presented in the manuscript we can learn several things about the model which can feedback later on model development as described in section 2.6 of the manuscript. The CAMS IFS model is just providing an example to show how this method can be applied efficiently in an operational forecasting system. It is also worth noting that the CAMS $CO_2$ forecast presented here is used by the scientific community for a variety of purposes (e.g. field experiments, boundary conditions). For this reason, we also think that the results, although specific to the CAMS $CO_2$ forecast model, could also be interesting to other scientists.

2. The applicability of this method to other systems is straightforward.

The method could be useful for any model to be used in forecast mode and suffering from substantial biases in their land ecosystem flux budget. The use of the method can be two-fold: as a bias correction to the land ecosystem fluxes or as a diagnostic of bias contribution from different regions/vegetation types. The system is flexible and cost-effective to run. It only needs a few components: (i) A reference budget which can be obtained from a climatology of optimized fluxes (e.g. the MACC product can be easily obtained from www-lscedods.cea.fr/invsat/ PYVAR14_MACC/V2/Fluxes/3Hourly and it is well documented); (ii) Past 10-day

[Figure]

NEE simulated by the forward model; (iii) The NEE anomaly of the forward model with respect to its climate based on a 10-year simulation. The use of the NEE anomaly is optional, and the benefits/drawbacks of using it will be described in the revised version of the paper (see further explanation in the minor comments).

3. Who or what profits from this improved $CO_2$ forecast?

The $CO_2$ forecast is a product freely available to the wide public and scientific community (http://atmosphere.copernicus.eu) with users from a variety of backgrounds. This will be emphasized in the revised version of the manuscript, including the main scientific research areas that can benefit from a $CO_2$ forecast which are listed below:

- **Global data assimilation of atmospheric $CO_2$ observations**

  The atmospheric $CO_2$ forecast is used as a prior to the atmospheric $CO_2$ analysis. For example, the CAMS atmospheric $CO_2$ analysis currently assimilates the GOSAT $CO_2$ product using a 4D-Var atmospheric data assimilation system (Massart et al. 2016). The reduction of the bias in the forecast by BFAS is highly desirable for data assimilation because the biases violate the assumption that the error distribution of the prior is centred around the true value.

  The $CO_2$ analysis system could be used to assimilate/combine a wide range of observations in the future. Preliminary monitoring/intercomparison of different $CO_2$ satellite products can be easily performed to provide feedback to the scientific community working on satellite retrievals. The fact that the forecast can provide a realistic representation of the underlying atmospheric variability of $CO_2$ in a timely manner is an important part of this data assimilation and monitoring processes. One of the most prominent modes of variability in the current 5-day forecast is the day-to-day synoptic variability. Thus, the emphasis is on synoptic timescales.

- **$CO_2$ observing system**
  The $CO_2$ forecast has been used in the research of bias corrections for satellite retrievals of OCO-2 lead by Chris O'Dell and could also be used in $CH_4$ satellite retrievals using the proxy method (Schepers et al. 2012). The predictive skill has also been used to support the planning of flight campaigns (e.g. CHARMEX, Ricaud et al. 2016, http://charmex.lsce.ipsl.fr/, and ACT-America, http://www-air.larc.nasa.gov/missions/ACT-America/) designed to improve our understanding of processes affecting atmospheric composition. It has also been used to demonstrate the use of new instruments in field experiments (e.g. Polarstern campaign, Klappenback et al. 2015). The detection of the atmospheric signals in the 1-day forecast (or nowcasting) can also help the interpretation of the observed variability from operational in situ networks (ICOS/InGOS monitoring), as well as expanding research networks (e.g. TCCON-RD) which aim to provide observations a few days behind real time.

- **$CO_2$ regional modelling**
  Another core usage of the global forecast is as boundary conditions for regional models. In particularly those studies focusing on city-scale resolution (e.g. Bréon et al. 2015, Boon et al. 2015) can benefit the most from the high resolution of the NWP global model.

Because of all these growing needs for a $CO_2$ analysis/forecast in real time, there have been recent efforts to start similar analysis/forecasting systems by NASA GMAO (http://acdb-ext.gsfc.nasa.gov/People/Colarco/Mission_Support/, Ott et al. 2015) and Environment Canada (Polavarapu et al. 2015) with their NWP models.

*\* Another issue with the paper is the choice of the control run. Taking the fluxes from the neutral-biosphere in CTESSEL is clearly wrong, and there could have been*

Interactive
comment

[Figure]

*many easy ways to improve on those. I think that a better benchmark is the available MACC fluxes, as the authors show that these already do quite a good job in matching observations if simply prescribed to the CAMS model. The authors state that these fluxes do not have synoptic variability, and I am not clear why this is because their resolution is never mentioned in the paper. But if diurnal and synoptic variations are needed, the simple method of Olsen and Randerson (2004) can be used to include the effect of temperature and light on monthly mean fluxes to get hourly ones. If the BAFS system was shown to be better than such an offline flux product, it would be much more clear to me that this way of BFAS is the way forward for CAMS.*

In the revised manuscript we will highlight the benefits of using BFAS to correct the modelled NEE as part of the CTESSEL land-surface model instead of using an offline flux product, e.g. the climatology of the MACC optimized fluxes (used as benchmark in the paper). The MACC optimized fluxes have a resolution of 3 hours, but all night-time and day-time variations for time scales less than a week only come from the underlying prior fluxes. Using a 10-year climatology means that the synoptic variability of the fluxes is not present. Agusti-Panareda et al (2014) showed that the synoptic variability of the fluxes could be important when it comes to represent the synoptic atmospheric $CO_2$ variability in the boundary layer. The Olsen and Randerson (2004) method could be used to remediate part of this problem. However, this solution would not be as straightforward to apply in an online forecast as it is done in an offline mode, for which all the climate forcing parameters (2 m temperature and solar radiation can be retrieved beforehand). There are also other reasons for not using an offline NEE product or optimized fluxes directly in the CAMS $CO_2$ forecasting system:

- Downscaling the coarse optimized fluxes (2.5x3.75 degrees) at the resolution used by NWP models (currently 9 km at ECMWF) is not straightforward. Inconsistencies in the topography (particularly around mountains and coastlines) makes the low resolution fluxes difficult to use in a high resolution model.

- Coupling of $CO_2$ fluxes from terrestrial vegetation and the atmospheric model represents an important step towards a better understanding of the interaction between the ecosystem and regional atmospheric processes (Lu et al. 2001, Moreira et al, 2013). Boussetta et al. (2013) showed that the coupling between the $CO_2$ fluxes and the water and energy fluxes in the modelling of vegetation can improve the simulation of surface parameters such as temperature and humidity as well as NEE. This coupling has been shown to benefit the simulation of the $CO_2$ diurnal cycle in the atmospheric boundary layer in the tropics (Lu et al., 2005, Moreira et al. 2013).

- Finally, because offline NEE products or optimized fluxes are not available in near-real time, we would need to use a climatology. The inter-annual variability associated with the land sink cannot be considered when using just a climatology of NEE. Despite being a challenging aspect of the modelling, we think it is worth having inter-annual variabilily in the model forecast. The main rationale for this is based on the understanding that the climate variables simulated in the NWP model – such as temperature and precipitation – play an important role in explaining the inter-annual variability of NEE (Schaefer et al. 2002). The motivation for including the model inter-annual variability in the flux adjustment will be clarified in the revised manuscript.

*\* It is not clear to me why certain metrics were chosen for evaluation. The authors present mean biases and standard deviations in Figures 9 and 10, correlation coefficients in Table 4, no metric for Figure 11, but there are never root- mean-square differences reported which I think are most useful. I think in figure 11 the MACC fluxes have the lowest RMSD than the BFAS fluxes. And from the captions it seems that both observations and simulations are done as daily (24-hour?) averages. I think that this daily averaging is needed because the independent adjustment of the GPP and TER scaling factors leads to strong variations in NEE that do not necessarily preserve a good diurnal cycle. But I might be wrong on that, as I could not assess this from the*

*figures shown. 24-hour average observations could have a lot of hour-to-hour variability which should be shows by an error bar. The statistics and figures moreover seem to cover only the month of March and a few selected days in March. It remains unexplained why this choice was made, and what the metrics look like for other months. I would expect for instance in summer to see even larger day-to-day variations in NEE, and then also in atmospheric $CO_2$*

Following the reviewer's advice, we have computed the root-mean-square (RMS) error of the different $CO_2$ experiments with respect to observations at the tower sites shown in Fig 11 of the manuscript (see Table 1 below). With the RMS error it is not as easy to see the improvement in the modelled variability as with the correlation coefficient $r$, because the RMS error increases very rapidly when there is large variability. This effect can be clearly seen at Park Falls at 30 m above the surface. Despite the substantial improvement in the model variability with BFAS ($r = 0.8$) compared to the CONTROL forecast ($r = 0.3$), the RMS error is larger in BFAS than in the CONTROL experiment by more than 1 ppm. This happens because the BFAS experiment overestimates the amplitude of the synoptic variability which is nearly non existent or even anticorrelated in the CONTROL experiment. At West Branch, the BFAS experiment has a much lower RMS error than both the experiments without BFAS and with optimized fluxes. Table 1 can be included in the supplement of the revised manuscript.

The impact of BFAS on the diurnal cycle amplitude has been evaluated in the northern hemisphere land (north of $20^oN$) based hourly data from all the in situ stations compiled in the NOAA Obspack (2015) dataset for 2010 (Fig. 1 of this reply). The mean error of the diurnal cycle amplitude (daily max value minus daily min value) is reduced for all seasons, with larger improvements in winter, autumn and spring. The RMS error on the other hand is slightly worsened. This is not surprising since the reference optimized flux dataset is not designed to represent the synoptic variability of the diurnal cycle amplitude (see green and dark blue bars in Fig. 1 of this reply). Summer months

**Table 1.** Root mean square error [ppm] of different forecast (FC) experiments with observations at three NOAA/ESRL tall towers for daily mean dry molar fraction of atmospheric $CO_2$ in March 2010. The dash symbol means the correlation is not significant.

| NOAA/ESRL Tower site (ID) | Latitude, Longitude, Altitude | Sampling level [m] | BFAS FC | CTRL FC | OPT FC | OPT-CLIM FC |
|---|---|---|---|---|---|---|
| Park Falls, Wisconsin (LEF) | $45.95^oN$, $90.27^oW$, 472 m | 30 122 396 | 6.12 4.05 2.93 | 4.97 5.44 5.10 | 3.04 2.09 1.37 | 3.31 3.06 1.99 |
| West Branch, Iowa (WBI) | $41.72^oN$, $91.35^oW$, 242 m | 31 99 379 | 3.79 2.91 2.46 | 10.39 9.94 8.91 | 5.06 2.95 3.20 | 6.96 3.92 2.43 |
| Argyle, Maine (AMT) | $45.03^oN$, $68.68^oW$, 50 m | 12 30 107 | 3.72 3.55 2.86 | 3.76 3.36 3.37 | 2.35 1.66 1.06 | 1.30 0.82 0.76 |

have larger diurnal cycle amplitudes and as expected the model also has larger errors in JJA. However, the impact of BFAS on the RMS error is the same for all months. This assessment of the diurnal cycle can be included in the supplement of the revised manuscript.

*  I would like to know what the added value is of having the gamma-parameter included in BFAS. The description of its calculation and adjustment is quite extensive but I do not really understand what role it plays. Perhaps there could be an experiment where BFAS is used without the adjustment in equation 3. After all, not needing the ensemble of forecasts would make the scheme a bit simpler, and perhaps just as good? I know I am likely to be wrong as the authors have decided to include this procedure in BFAS, but I would like to see the evidence to support that decision.*

A new experiment has been performed in which the $\gamma$ factor is set to zero in order to demonstrate the value of having the inter-annual variability in BFAS. Indeed the inter-annual variability can be important factor in the simulation of $CO_2$ (Schaefer et al. 2002, Chamard et al. 2003). However, because is not the same in every region/season/year it can also be difficult to demostrate its impact with observations (Figs 2, 3, 4 and 5 of this reply). In BFAS, the use of the $\gamma$ factor to represent the inter-annual variability from the model generally has a small impact. However, there are seasons and regions where we see the impact of using the $\gamma$ factor. As expected, this impact tends to be larger in the tropics, where the model inter-annual variability is also largest (Agusti-Panareda et al. 2014). However, we can also see some impact in the northern and sourthern hemisphere for the MAM, JJA, SON seasons. In summary, including the inter-annual variability factor in BFAS is beneficial as in most cases it leads to a bias reduction, with just a few exceptions for the SON season (see LN20N in Fig. 1 and LTrop in Fig. 4 of this reply).

[Figure]

**Minor comments**

\* *Page 3, line 5: I do not agree that the current monitoring of $CO_2$ relies on satellites and it is even a bit insulting to the real monitoring groups to say it. I suggest to change it because satellites do not yet see reliable $CO_2$. In fact, the second part of this statement is also not right because the observations you show and that MACC fluxes rely on mostly come from flasks and not from in-situ instruments.*

The reference to in situ observations was meant to include both continuous and flask measurements (lines 8 to 10 in Page 3). In the revised version of the manuscript this will be clarified by specifying both explicitly.

\* *Page 12, line 20: the current adjustment scheme for GPP and TER does not include any covariances between the adjustments, but we know that they often respond in the same direction and that errors are correlated. It would be good to think about an adjustment scheme that uses such information. Showing the posterior diurnal cycle is also needed.*

This will be mentioned as future improvements planned for BFAS in section 6.1 of the revised manuscript. The impact on the diurnal cycle will be included in the supplement as mentioned above.

\* *Page 13, line 20: You use now the names OPT-CLIM and later on in the text and tables CLIM-OPT. Is this the same run? It was to me confusing. Also see later remark about Table 2*

The runs are the same and the text will be corrected in the revised version.

\* *Page 14, line 20: A table listing the annual mean fluxes for transcom regions for all simulations would be valuable I think*

The proposed table for the budget in the Transcom regions will be included in the revised version.

*\* Page 15, line 25: The SH problems could come from a different north to south transport characteristic of the two atmospheric models used (IFS and LMDZ?). Can this be illustrated with a simple SF6 simulation and compare it to observations?*

We think the negative bias in the southern hemisphere comes from biases in tropical Africa. Preliminary experiments to assimilate IASI $CO_2$ using the $CO_2$ forecast have shown a large systematic difference throughout the free tropospheric column over tropical Africa which is consistent with the negative bias in the southern hemisphere. This will be mentioned in the revised manuscript.

*\* Acknowledgements: please check the data usage policy of NOAA as I do not believe you can simply take data from their FTP and then publish it with this statement.*

The authors have contacted Ed Dlugokencky regarding the acknowledgements and received his confirmation that these are sufficient. An acknowledgement for the Obspack data used for the plots in the supplement of the revised manuscript will be added.

*\* Page 30, Table 2: I was confused because it says that CLIM-OPT uses MACC fluxes as reference in BFAS but from the methods I understood that CLIM-OPT or OPT-CLIM used the climatological fluxes from MACC directly as underlying biosphere fluxes? I discovered this only towards the end of reading and it made me think I misunderstood the simulations completely. Even now I doubt it.*

CLIM-OPT uses the climatological fluxes from MACC (i.e. the total $CO_2$ flux) and BFAS justs uses a climatology of the MACC residual biosphere fluxes. This will be clarified in Table 2 and in the text of the revised manuscript.

*\* Figures 4 and 7: it would be better to use PgC/yr as units and not GtC/day because now they just look very small on the y-axis with many insignificant digits to start.*

The units will be changed in the revised version of the manuscript.

* *I believe Figure 12 and 13 are not needed and could be removed.*

The authors disagree on this point. The fact that BFAS can change the gradient of the fluxes and as a result improve the atmospheric $CO_2$ synoptic variability is an achievement that needs to be properly documented.

**REFERENCES**

A. Agusti-Panareda, S. Massart, F. Chevallier and S. Boussetta, G. Balsamo, A. Beljaars and P. Ciais, N.M. Deutscher, R. Engelen and L. Jones, R. Kivi, J.-D. Paris, V.-H. Peuch and V. Sherlock, A.T. Vermeulen, P.O. Wennberg and D. Wunch: Forecasting global atmospheric $CO_2$, *Atmospheric Chemistry and Physics*, 14, 11959-11983, doi:10.5194/acp-14-11959-2014, 2014.

Boon, A. and Broquet, G. and Clifford, D. J. and Chevallier, F. and Butterfield, D. M. and Pison, I. and Ramonet, M. and Paris, J. D. and Ciais, P. (2015): TITLE = Analysis of the potential of near ground measurements of $CO_2$ and $CH_4$ in London, UK for the monitoring of city-scale emissions using an atmospheric transport model, *Atmospheric Chemistry and Physics Discussions*, 15, 33003–33048, doi:10.5194/acpd-15-33003-2015.

Boussetta, S., Balsamo, G., Beljaars, A., Agusti-Panareda, A., Calvet, J.-C., Jacobs, C., van den Hurk, B., Viterbo, P., Lafont, S., Dutra, E., Jarlan, L., Balzarolo, M., Papale, D., and van der Werf, G. (2013) : Natural carbon dioxide exchanges in the ECMWF Integrated Forecasting System: implementation and offline validation, *J. Geophys. Res.-Atmos.*, 118, 1–24, doi: 10.1002/jgrd.50488.

Bréon, F. M. and Broquet, G. and Puygrenier, V. and Chevallier, F. and Xueref-Remy,

I. and Ramonet, M. and Dieudonné, E. and Lopez, M. and Schmidt, M. and Perrussel, O. and Ciais, P. (2015): An attempt at estimating Paris area $CO_2$ emissions from atmospheric concentration measurements, *Atmospheric Chemistry and Physics*, 15, 1707–1724, doi: 10.5194/acp-15-1707-2015.

Chamard, P., F. Thiery, A. Di Sarra, L. Ciattaglia, L. De Silvestri, P. Grigioni, F. Monteleone and S. Piacentino (2003): Inter-Annual variability of atmospheric CO2 in the Mediterranean: measurements at the island of Lampedusa. *Tellus B*, 55: 83–93. doi: 10.1034/j.1600-0889.2003.00048.x.

Chen, Z. H., Zhu, J. and Zeng, N. (2013): Improved simulation of regional $CO_2$ surface concentrations using GEOS-Chem and fluxes from VEGAS, *Atmospheric Chemistry and Physics*, 13 , 7607–7618, 10.5194/acp-13-7607-2013.

Chevallier F., N.M. Deutscher, T.J. Conway, P. Ciais, L. Ciattaglia, S. Dohe, M. Fröhlich, A.J. Gomez-Pelaez , D. Griffith, F. Hase, L. Haszpra, P. Krummel, E. Kyrö, C. Labuschne, R. Langenfelds, T. Machida, F. Maignan, H. Matsueda , I. Morino, J. Notholt, M. Ramonet, Y. Sawa , M. Schmidt, V. Sherlock, P. Steele, K. Strong , R. Sussmann, P. Wennberg, S. Wofsy, D. Worthy , D. Wunch, M. Zimnoch (2011): Global $CO_2$ fluxes inferred from surface air-sample measurements and from TCCON retrievals of the $CO_2$ total column, *Geophys. Res. Let.*, 38, doi:10.1029/2011GL049899.

Chevallier, F. (2015): On the statistical optimality of $CO_2$ atmospheric inversions assimilating $CO_2$ column retrievals, *Atmospheric Chemistry and Physics*, 15, 11133–11145, doi:10.5194/acp-15-11133-2015, http://www.atmos-chem-phys.net/15/11133/2015/.

Keppel-Aleks,G. , P.O. Wennberg , R.A. Washenfelder and D. Wunch , T. Schneider , G.C. Toon , R.J. Andres and J.-F. Blavier , B. Connor , K.J. Davis , A.R. Desai and J. Messerschmidt , J. Notholt , C. M. Roehl , V. Sherlock , B.B. Stephens , S.A. Vay , S. C. Wofsy (2012): The imprint of surface fluxes and transport on variations in total column carbon dioxide, Biogeosciences, 9, 875-891, doi:10.5194/bg-9-875-2012.
Klappenbach, F., Bertleff, M., Kostinek, J., Hase, F., Blumenstock, T., Agusti-Panareda, A., Razinger, M., Butz, A. (2015): Accurate mobile remote sensing of $XCO_2$ and $XCH_4$ latitudinal transects from aboard a research vessel, *Atmospheric Measurement Techniques*, 8, 5023–5038, doi:10.5194/amt-8-5023-2015, 2015.

Lu, L., A.S. Denning, M.A. da Silva Dias, P. Silva-Dias, M. Longo, S.R. Freitas, and S. Saatchi (2005): Mesoscale circulation and atmospheric CO2 variation in the Tapajos Region, Para, Brazil. *Journal of Geophysical Research*, 110, D21102, doi:10.1029/2004JD005757.

Lu, L., R.A.Pielke, Sr., G.E. Liston, W. Parton, D. Ojima, and M. Hartman (2001): The Implementation of a two-way Interactive Atmospheric and Ecological Model and its Application to the Central United States. *Journal of Climate*, 14, 900–919.

Massart, S., Agustí-Panareda, A., Heymann, J., Buchwitz, M., Chevallier, F., Reuter, M., Hilker, M., Burrows, J. P., Deutscher, N. M., Feist, D. G., Hase, F., Sussmann, R., Desmet, F., Dubey, M. K., Griffith, D. W. T., Kivi, R., Petri, C., Schneider, M., Velazco, V. A. (2016): Ability of the 4-D-Var analysis of the GOSAT BESD $XCO_2$ retrievals to characterize atmospheric $CO_2$ at large and synoptic scales, *Atmospheric Chemistry and Physics*, 16, 1653–1671, doi:10.5194/acp-16-1653-2016.

Messerschmidt, J., N. Parazoo, D. Wunch, N.M. Deutscher, C. Roehl, T. Warneke and P.O.Wennberg (2013): Evaluation of seasonal atmosphere-biosphere exchange estimations with TCCON measurements, *Atmospheric Chemistry and Physics*, 13 , 5103-5115, doi:10.5194/acp-13-5103-2013.

Moreira, D. S., Freitas, S. R., Bonatti, J. P., Mercado, L. M., Rosário, N. M. É., Longo, K. M., Miller, J. B., Gloor, M., Gatti, L. V. (2013): Coupling between the JULES land-surface scheme and the CCATT-BRAMS atmospheric chemistry model (JULES-CCATT-BRAMS1.0): applications to numerical weather forecasting and the $CO_2$ budget in South America, *Geoscientific Model Development*, 6, 1243–1259, doi:10.5194/gmd-6-1243-2013.

Nassar, R., Jones, D. B. A., Suntharalingam, P., Chen, J. M., Andres, R. J., Wecht, K. J., Yantosca, R. M., Kulawik, S. S., Bowman, K. W., Worden, J. R., Machida, T., Matsueda, H. (2010): Modeling global atmospheric $CO_2$ with improved emission inventories and $CO_2$ production from the oxidation of other carbon species, *Geosci. Model Dev.*, 3, 689–716, doi:10.5194/gmd-3-689-2010.

NOAA Obspack (2015): Cooperative Global Atmospheric Data Integration Project; Multi-laboratory compilation of atmospheric carbon dioxide data for the period 1968-2014; `obspack_co2_1_GLOBALVIEWplus_v1.0_2015-07-30`; NOAA Earth System Research Laboratory, Global Monitoring Division. doi: 10.15138/G3RP42, http://dx.doi.org/10.15138/G3RP42

Olsen, S.C. and Randerson,J.T. (2004): Differences between surface and column atmospheric $CO_2$ and implications for carbon cycle research, *Journal of Geophysical Research*, 109, D02301, doi:10.1029/2003JD003968.

Polavarapu, S.M., M. Neish, M. Tanguay, C. Girard, J. de Grandpré, S. Gravel, K. Semeniuk, and D. Chan (2015): Adapting a weather forecast model for greenhouse gas simulation, *AGU Fall Metting*, San Francisco, 15-18 December 2015, Poster A31B-0037.

P. Ricaud, R. Zbinden, V. Catoire, V. Brocchi, F. Dulac, E. Hamonou, J.-C. Canonici, L. El Amraoui, S. Massart, B. Piguet, U. Dayan, P. Nabat, J. Sciare, M. Ramonet, M. Delmotte, A. G. di Sarra, D. Sferlazzo, T. Di Iorio, S. Piacentino, P. Cristofanelli, N. Mihalopoulos, G. Kouvarakis, S. Kleanthous, M. Pikridas, C. Savvides, R. E. Mamouri, A. Nisantzi, D. G. Hadjimitsis, J.-L. Attié, H. Ferré, P. Theron, Y. Kangah, N. Jaidan, P. Jacquet, S. Chevrier, C. Robert, A. Bourdon, J.-F. Bourdinot, and J.-C. Etienne (2016): Overview of the Gradient in Longitude of Atmospheric constituents above the Mediterranean basin (GLAM) airborne summer campaign (submitted to ACP).

MS No.: acp-2016-295 MS Type: Research article Iteration: Initial Submission Special

Schaefer, K., A.S. Denning, N. Suits, J.Kaduk, I. Baker, S. Los, L. Prihodko (2002): Effect of climate on inter-annual variability of terrestrial $CO_2$ fluxes, *Global Biogeo-chemical Cycles*, 16.

Schepers, D, S. Guerlet, A. Butz and J.Landgraf, C. Frankenberg, O. Hasekamp and J.-F. Blavier, N.M. Deutscher, D.W.T. Griffith , F. Hase, E. Kyro, I. Morino and V. Sherlock, R. Sussmann, I. Aben (2012): Methane retrievals from Greenhouse Gases Observing Satellite (GOSAT) shortwave infrared measurements: Performance comparison of proxy and physics retrieval algorithms, *Journal of Geophysical Research*, 117, D10307, doi:10.1029/2012JD017549.

Sweeney, C., A. Karion, S. Wolter, T. Newberger, D. Guenther, J. A. Higgs, A. E. Andrews, P. M. Lang, D. Neff, E. Dlugokencky, J. B. Miller, S. A. Montzka, B. R. Miller, K. A. Masarie, S. C. Biraud, P. C. Novelli, M. Crotwell, A. M. Crotwell, K. Thoning, and P. P. Tans (2015): Seasonal climatology of CO2 across North America from aircraft measurements in the NOAA/ESRL Global Greenhouse Gas Reference Network, *Journal of Geophysical Research: Atmospheres*, 120, 10, doi:10.1002/2014JD022591.

**FULL FIGURE CAPTIONS**

*Figure 1: Evaluation of diurnal cycle amplitude of $CO_2$ dry molar mixing ratio [ppm] for the different forecast experiments (see legend) in the northern hemisphere land (north of 20°N) based on hourly data from all the in situ stations compiled in the NOAA Obspack (2015) dataset for 2010. Top panel: mean error; middle panel: root mean square error; and lower panel: number of observations.*

*Figure 2: Mean error of atmospheric $CO_2$ dry molar mixing ratio [ppm] for different fore-*

[Figure]

*cast experiments (see legend) with respect to insitu and flask observations for different seasons and regions (N20N: north of 20°N; Trop: between 20°S and 20°N; S20S : south of 20°S) with a separation between land and sea points denoted by a preceeding "L" and "S" in the region name respectively. The observations were extracted from the NOAA Obspack (2015) dataset in 2010. The number of observations used for the statistics are shown as grey bars in the panel below each plot.*

*Figure 3: Root mean square error of atmospheric $CO_2$ dry molar mixing ratio [ppm] for different experiments (see legend) with respect to insitu and flask observations for different seasons and regions as described in Fig. 2. The observations were extracted from the NOAA Obspack (2015) dataset in 2010. The number of observations used for the statistics are shown as grey bars in the panel below each plot.*

*Figure 4: Mean error of atmospheric $CO_2$ dry molar mixing ratio [ppm] for different experiments (see legend) with respect to NOAA aircraft vertical profiles (Sweeney et al. 2015) in the free troposphere (1000 m above surface) for different seasons and regions as described in Fig. 2. The observations were extracted from the NOAA Obspack (2015) dataset in 2010. The number of observations used for the statistics are shown as grey bars in the panel below each plot.*

*Figure 5: Root mean square error of atmospheric $CO_2$ dry molar mixing ratio [ppm] for different experiments (see legend) with respect to NOAA aircraft vertical profiles (Sweeney et al. 2015) in the free troposphere (1000 m above surface) for different seasons and regions as described in Fig. 2. The observations were extracted from the NOAA Obspack (2015) dataset in 2010. The number of observations used for the statistics are shown as grey bars.*

[Figure]

[Figure]

**Fig. 1.** Evaluation of diurnal cycle amplitude of $CO_2$ [ppm] for the different forecast experiments (legend) in the NH land based on data from in situ stations (NOAA Obspack 2015). See full caption in text.

**Fig. 2.** Mean error of atmospheric CO2 [ppm] for different forecast experiments (see legend) with respect to insitu and flask observations from NOAA Obspack (2015). See full caption in text.

[Figure]

**Fig. 3.** RMS error of atmospheric CO2 [ppm] for different experiments (see legend) with respect to insitu and flask observations from NOAA Obspack (2015). See full caption in text.

[Figure]

**Fig. 4.** Mean error of atmospheric CO2 [ppm] for different experiments (see legend) with respect to NOAA aircraft vertical profiles (Sweeney et al. 2015) in the free troposphere. See full caption in text.

[Figure]

**Fig. 5.** RMS error of atmospheric CO2 [ppm] for different experiments (see legend) with respect to NOAA aircraft vertical profiles (Sweeney et al. 2015) in the free troposphere. See full caption in text.

---

## Author Comment (AC2) · 6 May 2016

**General comments**

- *This paper presents an enhancement to the CO2 assimilation system used within the Copernicus tracer assimilation system at ECMWF. The enhancement is certainly useful and potentially quite important but it comes with its own problems. I believe these need to be discussed in the manuscript and addressed in how the new product is made available.*

[Figure]

We thank the reviewer for his insightful comments concerning the potential use of the CAMS $CO_2$ analysis product in flux inversion systems. The reply to each of the reviewer's points can be found below.

- *The enhancement addresses the problem of large-scale biases in the fluxes which underlie the prior concentrations used in the assimilation. These biases are a serious matter since they mean that the probability densities assumed in the assimilation system (centered on the true value) don't, in fact, hold. So this is a potentially valuable improvement.*

This is a very important part of the motivation of this work because the atmospheric $CO_2$ forecast provides the prior information to the CAMS atmospheric $CO_2$ data assimilation. As the reviewer points out, the data assimilation system is only designed to reduce the random error, not the bias. Therefore, it is very important to bias correct the prior atmospheric mixing ratios from the forecast before assimilating any $CO_2$ observations. We will include this point in the introduction of the revised manuscript to strengthen the motivation for BFAS.

- *The problem arises when we consider what the generated CO2 fields are used for. Although there is probably some benefit for improved retrievals of temperature and moisture by improving the CO2 field the overwhelming use for the assimilated CO2 products is in estimating surface fluxes. the statistical apparatus is identical to the assimilation of the CO2 fields and the same restrictions apply. Among them is a firm prohibition on reusing information and the requirement that observations and prior are independent. Both of these are potentially violated in any downstream use of the BFAS product. Let's deal with these two problems in turn.*

The reviewer has an important point in that users of the CAMS $CO_2$ analysis/forecast products need to know what is the input data going into the product and what is the final uncertainty of the product. This is the case whether the users are working on flux inversion systems, planning of field experiments or using the product as boundary
conditions for regional models.

First of all, information on the uncertainty of the atmospheric $CO_2$ forecast with and without BFAS compared to the optimized flux experiments will be provided in terms of bias and root mean square error (RMSE) for different regions/seasons in the supplement of the revised manuscript using barplots as shown in Figures 1, 2, 3 and 4 in this reply.

Regarding the mixing of information in the analysis, this is currently not an issue for the CAMS $CO_2$ analysis system because the optimized fluxes used in BFAS are not based on satellite products; whereas the CAMS atmospheric $CO_2$ analysis is currently only assimilating satellite products. This will be clarified in the revised manuscript.

For the users, we envisage that the atmospheric $CO_2$ analysis/forecast will be used as boundary conditions for regional flux inversion systems. In this case the possible correlated errors between such an analysis and the measurements assimilated by the inversion within the regional domain will likely be marginal, given all the processing that is involved between the inversion to estimate the MACC optimized fluxes, BFAS and the IFS 4D-Var used by the CAMS atmospheric $CO_2$ analysis. The possibility to infer the surface fluxes directly from the IFS $CO_2$ analysis would mean that some information from the observations assimilated by the MACC flux inversion system would already be present in the CAMS $CO_2$ analysis via BFAS. Thus, we will detail the used observations in the revised manuscript.

• *I believe this paper is a potentially valuable contribution and look forward to the authors' revision. If the authors accept my first point about the mixing of data into their CO2 field then they also need to find a way of detailing which data was used to generate the flux fields that underlie BFAS.*

The flux fields underlying BFAS are primarily NEE modelled by the CTESSEL Carbon module in the IFS (Boussetta et al. 2013), which are then re-scaled using continental-scale climatological budgets from the MACC optimized fluxes of Chevallier et al. (2011,

2015). There is also some input from the EDGAR v4.2 anthropogenic emissions and the biomass burning emissions from GFAS (Kaiser et al. 2012). The information from these inventories is used to extract the NEE as a residual from the optimized fluxes.

The documentation of the different data streams going into BFAS and their access (via the Copernicus Data Catalogue and the EDGAR database) will be detailed in the supplement of the revised manuscript.

**Specific comments**

• *The assimilated CO2 field now includes information from a prior informed by a previous flux inversion. This inversion presumably used measurements from the in situ network, aircraft and/or TCCON. We can't tell which without a detailed examination of the papers that underlie that inversion. We need to know because, if we're going to use the BFAS product to drive a future inversion, we need to exclude those measurements. One might argue that the periods don't overlap but the evidence of the paper shows that the model-data mismatch is so strongly correlated from year to year (consistent seasonal errors in the pre-BFAS version) that this doesn't avoid the problem.*

In the revised manuscript we will mention that since the BFAS product contains information from the optimized fluxes, users should be aware that the optimized fluxes assimilated most available background air-sample monitoring sites (listed in the supplement of Chevallier et al. 2015, see http://www.atmos-chem-phys.net/15/11133/2015/acp-15-11133-2015-supplement.pdf.

Although we expect that observations ingested by the MACC inversion system of Chevallier et al (2011, 2015) will have an influence on the BFAS fluxes to some extent, we cannot quantify their degree of influence in this paper. We expect some information from the observations will be lost in the flux inversion process and specially in BFAS.

The processing in BFAS involves spatial/temporal smoothing of the optimized fluxes over land with a 10-year averaging to construct the climatology and then the inclusion of the model interannual variability. The influence from these surface observations will be further diminished after the assimilation of satellite products in the analysis. In order to ensure independence between the $CO_2$ analysis and the background-air observations ingested by the MACC inversion system, the atmospheric $CO_2$ analysis could be sampled at non-background-air locations characterized by a large influence from the satellite products.

• *The second problem, of the prior estimate for a flux inversion being partially reflected in the data we use is not new with BFAS. It exists in the original Copernicus products too. I'm unsure whether the mixing data and model information in the prior CO2 field makes this problem worse but it seems like it should.*

The BFAS processing should bring the mean error and large-scale spatial distribution of the CTESSEL NEE fluxes closer to the MACC optimized fluxes. This probably implies that the BFAS fluxes will not be completely independent from the prior in the MACC flux inversion system. Thus, if the same prior would be used again to infer fluxes from the atmospheric $CO_2$ analysis data, then it would be likely that BFAS would make the problem associated with their lack of independence worse.

• *Finally there is the question of the uncertainty of the BFAS CO2 field. There are two countervaling effects in play. First the bias correction of the prior has reduced residuals in the generated CO2 field so that uncertainties (which are the statistics of the difference between estimated and true values) seem to have reduced. On the other hand an extra process has been added to the assimilation with a new set of parameters to scale prior fluxes. These will have their own uncertainty and should (since the posterior CO2 field is sensitive to its prior) increase posterior uncertainty. Which of these wins out? I am always a little wary of criticizing a paper for things it did not do since no piece of research is complete. However it's an important general rule that products that are to*

*be used as inputs to statistical procedures such as flux inversions need to specify their
uncertainty as well as their mean.*

Plots showing characteristic biases and root mean square errors of the BFAS $CO_2$ field
for different seasons/regions will be included in the supplement of manuscript (see
Figures below). These plots use all the observations from the NOAA Obspack (2015)
dataset (excluding only the observations from CONTRAIL and HIPPO flights).

**REFERENCES**

Boussetta, S., Balsamo, G., Beljaars, A., Agusti-Panareda, A., Calvet, J.-C., Jacobs, C.,
van den Hurk, B., Viterbo, P., Lafont, S., Dutra, E., Jarlan, L., Balzarolo, M., Papale, D.,
and van der Werf, G. (2013) : Natural carbon dioxide exchanges in the ECMWF Inte-
grated Forecasting System: implementation and offline validation, *J. Geophys. Res.-
Atmos.*, 118, 1–24, doi: 10.1002/jgrd.50488.

Chevallier F., N.M. Deutscher, T.J. Conway, P. Ciais, L. Ciattaglia, S. Dohe, M. Fröh-
lich, A.J. Gomez-Pelaez , D. Griffith, F. Hase, L. Haszpra, P. Krummel, E. Kyrö,
C. Labuschne, R. Langenfelds, T. Machida, F. Maignan, H. Matsueda , I. Morino,
J. Notholt, M. Ramonet, Y. Sawa , M. Schmidt, V. Sherlock, P. Steele, K. Strong
, R. Sussmann, P. Wennberg, S. Wofsy, D. Worthy , D. Wunch, M. Zimnoch
(2011): Global CO2 fluxes inferred from surface air-sample measurements and
from TCCON retrievals of the CO2 total column, *Geophys. Res. Let.*, 38,
doi:10.1029/2011GL049899.

Chevallier, F. (2015): On the statistical optimality of $CO_2$ atmospheric inversions assim-
ilating $CO_2$ column retrievals, *Atmospheric Chemistry and Physics*, 15, 11133–11145,
doi:10.5194/acp-15-11133-2015, http://www.atmos-chem-phys.net/15/11133/2015/.

Kaiser, J.K., A. Heil, M.O. Andreae, A. Benedetti, N. Chubarova, L. Jones, J.-J. Morcrette, M. Razinger, M.G. Schultz, M. Suttie, G.R. van der Werf (2012): Biomass burning emissions estimated with a global fire assimilation system based on observed fire radiative power, Biogeosci., 9, 527–554, doi:10.5194/bg-9-527-2012.

NOAA Obspack (2015): Cooperative Global Atmospheric Data Integration Project; Multi-laboratory compilation of atmospheric carbon dioxide data for the period 1968-2014;
`obspack_co2_1_GLOBALVIEWplus_v1.0_2015-07-30`; NOAA Earth System Research Laboratory, Global Monitoring Division. doi: 10.15138/G3RP42, http://dx.doi.org/10.15138/G3RP42

Sweeney, C., A. Karion, S. Wolter, T. Newberger, D. Guenther, J. A. Higgs, A. E. Andrews, P. M. Lang, D. Neff, E. Dlugokencky, J. B. Miller, S. A. Montzka, B. R. Miller, K. A. Masarie, S. C. Biraud, P. C. Novelli, M. Crotwell, A. M. Crotwell, K. Thoning, and P. P. Tans (2015): Seasonal climatology of CO2 across North America from aircraft measurements in the NOAA/ESRL Global Greenhouse Gas Reference Network, *Journal of Geophysical Research: Atmospheres*, 120, 10, doi:10.1002/2014JD022591.

**FULL FIGURE CAPTIONS**

*Figure 1: Mean error of atmospheric $CO_2$ dry molar mixing ratio [ppm] for different forecast experiments (see legend) with respect to insitu and flask observations for different seasons and regions (N20N: north of $20^o$N; Trop: between $20^o$S and $20^o$N; S20S : south of $20^o$S) with a separation between land and sea points denoted by a preceeding "L" and "S" in the region name respectively. The observations were extracted from the NOAA Obspack (2015) dataset in 2010. The number of observations used for the statistics are shown as grey bars in the panel below each plot.*

*Figure 2: Root mean square error of atmospheric $CO_2$ dry molar mixing ratio [ppm] for different experiments (see legend) with respect to insitu and flask observations for*

*different seasons and regions as described in Fig. 1. The observations were extracted from the NOAA Obspack (2015) dataset in 2010. The number of observations used for the statistics are shown as grey bars in the panel below each plot.*

*Figure 3: Mean error of atmospheric $CO_2$ dry molar mixing ratio [ppm] for different experiments (see legend) with respect to NOAA aircraft vertical profiles (Sweeney et al. 2015) in the free troposphere (1000 m above surface) for different seasons and regions as described in Fig. 1. The observations were extracted from the NOAA Obspack (2015) dataset in 2010. The number of observations used for the statistics are shown as grey bars in the panel below each plot.*

*Figure 4: Root mean square error of atmospheric $CO_2$ dry molar mixing ratio [ppm] for different experiments (see legend) with respect to NOAA aircraft vertical profiles (Sweeney et al. 2015) in the free troposphere (1000 m above surface) for different seasons and regions as described in Fig. 1. The observations were extracted from the NOAA Obspack (2015) dataset in 2010. The number of observations used for the statistics are shown as grey bars.*
* * *
[Figure]

**Fig. 1.** See main text for full figure caption.

**Fig. 2.** See main text for full figure caption.

[Figure]

[Figure]

**Fig. 3.** See main text for full figure caption.

[Figure]

**Fig. 4.** See main text for full figure caption.

---

## Author Response (AR1)

**Reply to the interactive comment from the reviewers on "A biogenic $CO_2$ flux adjustment scheme for the mitigation of large-scale biases in global atmospheric $CO_2$ analyses and forecasts" by Agustí-Panareda et al.**

**Reply to reviewer 1**

We thank the reviewer for his/her comments. We have taken them into account in the revised manuscript to improve the motivation and the message of this work. In particular, we have highlighted the scientific content of our results. In the reply below we address all the reviewer's concerns in order to clarify any misunderstanding on the importance of this study, and its relevance for the scientific community working on atmospheric composition and the carbon cycle. A pointer to the the different parts of the paper that have been modified is also provided in blue text for each general and specific comment addressed. The modifications performed in the revised paper are also highlighted in the latexdiff file provided.

**General comments**

* *In my opinion this paper has a number of problems and I believe that it is not currently suitable for publication in ACP. The first is that the paper contains relatively little scientific content, and there is nearly nothing that can be learned from the paper for a big audience. And even for researchers in the field of atmospheric $CO_2$ modeling, these methods are very system specific and not easily used by others even if they needed such flux adjustments. So this paper should probably remain a technical report for the Copernicus project, or perhaps it can be published in Geophysical Model Development journal. The case of why having better synoptic variations in forecast $CO_2$ is important is also not clearly made I think: who or what profits from this improved $CO_2$ forecast?*

The major aspects raised by the reviewer are addressed separately in detail below:

1. The scientific content of the paper.

   Any atmospheric $CO_2$ forecast system requires a flux adjustment of some sort in order to constrain the budget of sources/sinks at the surface and avoid the growth of biases in the atmospheric background as documented by Agusti-Panareda et al. (2014). The scientific question addressed in this paper is how to use the best information we have in near-real time to adjust the fluxes in a way that reduces the bias of the atmospheric $CO_2$ forecast with the minimum deterioration of the synoptic skill. The simple flux adjustment scheme proposed here is based on a climatology of optimized fluxes and it could be applied

easily to other models. In the past other methods have been used by several modelling studies to remove biases attributed to the NEE fluxes. For instance, by globally re-scaling balanced NEE fluxes to match the residual land sink given by a climatology of TRANSCOM optimized fluxes (Nassar et al., 2010; Chen et al.,2013), or by re-scaling locally the NEE at boreal regions in order to get a better fit in the seasonal cycle (e.g. Messerschmidt et al. 2013, Keppel-Aleks et al. 2012).

This paper addresses the challenge of designing an online bias correction in a forecasting system with the aim to deliver an atmospheric $CO_2$ forecast and analysis that can be useful to the scientific community. The other methods mentioned previously are designed to work as a one-off correction and they offer less flexibility because they are performed offline. Tunning model parameters and/or re-scaling fluxes offline are not sufficient to garantee a bias reduction in the system. An online adaptive system is required because errors in the meteorology can evolve as a result of regular operational Numerical Weather Prediction model upgrades and these affect the NEE budget in the model.

**An extract of the paragraphs above have been included in the methodology section.**

From the flux adjustment method presented in the manuscript we can learn several things about the model which can feedback later on model development as described in section 2.6 of the manuscript. The CAMS IFS model is just providing an example to show how this method can be applied efficiently in an operational forecasting system. It is also worth noting that the CAMS $CO_2$ forecast presented here is used by the scientific community for a variety of purposes (e.g. field experiments, boundary conditions). For this reason, we also think that the results, although specific to the CAMS $CO_2$ forecast model, could also be interesting to other scientists.

**An extract of the paragraph above has been included in the new Discussion subsection entitled "Aspects to be considered by users" as well as the summary section.**

2. The applicability of this method to other systems is straightforward.

The method could be useful for any model to be used in forecast mode and suffering from substantial biases in their land ecosystem flux budget. The use of the method can be two-fold: as a bias correction to the land ecosystem fluxes or as a diagnostic of bias contribution from different regions/vegetation types. The system is flexible and cheap to run. It only needs a few components: (i) A reference budget which can be obtained from a climatology of optimized fluxes (e.g. the MACC product can be easily obtained from `www-lscedods.cea.fr/invsat/PYVAR14_MACC/V2/Fluxes/3Hourly` and it is well documented); (ii) Past 10-day NEE simulated by the forward model; (iii) The NEE anomaly of the forward model with respect to its climate based on a 10-year simulation. The use of the NEE anomaly is optional, and the benefits/drawbacks of using it are described in the revised version of the paper (see further explanation in the minor comments).

**An extract of the paragraph above has been included in the discussion section (first paragraph).**

3. Who or what profits from this improved $CO_2$ forecast?

The $CO_2$ forecast is a product freely available to the wide public and scientific community (`http://atmosphere.copernicus.eu`) with users from a variety of backgrounds. This

will be emphasized in the revised version of the manuscript, including the main scientific research areas that can benefit from a $CO_2$ forecast which are listed below:

- **Global data assimilation of atmospheric $CO_2$ observations**

  The atmospheric $CO_2$ forecast is used as a prior to the atmospheric $CO_2$ analysis. For example, the CAMS atmospheric $CO_2$ analysis currently assimilates the GOSAT $CO_2$ product using a 4D-Var atmospheric data assimilation system (Massart et al. 2016). The reduction of the bias in the forecast by BFAS is highly desirable for data assimilation because the biases violate the assumption that the error distribution of the prior is centred around the true value.

  The $CO_2$ analysis system could be used to assimilate/combine a wide range of observations in the future. Preliminary monitoring/intercomparison of different $CO_2$ satellite products can be easily performed to provide feedback to the scientific community working on satellite retrievals. The fact that the forecast can provide a realistic representation of the underlying atmospheric variability of $CO_2$ in a timely manner is an important part of this data assimilation and monitoring processes. One of the most prominent modes of variability in the current 5-day forecast is the day-to-day synoptic variability. Thus, the emphasis is on synoptic timescales.

- **$CO_2$ observing system**

  The $CO_2$ forecast has been used in the research of bias corrections for satellite retrievals of OCO-2 lead by Chris O'Dell and could also be used in $CH_4$ satellite retrievals using the proxy method (Schepers et al. 2012). The predictive skill has also been used to support the planning of flight campaigns (e.g. CHARMEX, Ricaud et al. 2016, `http://charmex.lsce.ipsl.fr/`, and ACT-America, `http://www-air.larc.nasa.gov/missions/ACT-America/`) designed to improve our understanding of processes affecting atmospheric composition. It has also been used to demonstrate the use of new instruments in field experiments (e.g. Polarstern campaign, Klappenback et al. 2015). The detection of the atmospheric signals in the 1-day forecast (or nowcasting) can also help the interpretation of the observed variability from operational in situ networks (ICOS/InGOS monitoring), as well as expanding research networks (e.g. TCCON-RD) which aim to provide observations a few days behind real time.

- **$CO_2$ regional modelling**

  Another core usage of the global forecast is as boundary conditions for regional models. In particularly those studies focusing on city-scale resolution (e.g. Bréon et al. 2015, Boon et al. 2015) can benefit the most from the high resolution of the Numerical Weather Prediction (NWP) global model.

Because of all these growing needs for a $CO_2$ analysis/forecast in real time, there have been recent efforts to start similar analysis/forecasting systems by NASA GMAO (`http://acdb-ext.gsfc.nasa.gov/People/Colarco/Mission_Support/`, Ott et al. 2015) and Environment Canada (Polavarapu et al. 2015) with their NWP models.

**The benefits of the improved forecast for the scientific community (data assimilation, CO2 observing system, and CO2 regional modellin) have been highlighted in the introduction and summary sections.**

*   *Another issue with the paper is the choice of the control run. Taking the fluxes from the neutral-biosphere in CTESSEL is clearly wrong, and there could have been many easy ways to improve on those. I think that a better benchmark is the available MACC fluxes, as the authors show that these already do quite a good job in matching observations if simply prescribed to the CAMS model. The authors state that these fluxes do not have synoptic variability, and I am not clear why this is because their resolution is never mentioned in the paper. But if diurnal and synoptic variations are needed, the simple method of Olsen and Randerson (2004) can be used to include the effect of temperature and light on monthly mean fluxes to get hourly ones. If the BAFS system was shown to be better than such an offline flux product, it would be much more clear to me that this way of BFAS is the way forward for CAMS.*

In the revised manuscript we have highlighted the benefits of using BFAS to correct the modelled NEE as part of the CTESSEL land-surface model instead of using an offline flux product, e.g. the climatology of the MACC optimized fluxes (used as benchmark in the paper). The MACC optimized fluxes have a resolution of 3 hours, but all night-time and day-time variations for time scales less than a week only come from the underlying prior fluxes. Using a 10-year climatology means that the synoptic variability of the fluxes is not present. Agusti-Panareda et al (2014) showed that the synoptic variability of the fluxes could be important when it comes to represent the synoptic atmospheric $CO_2$ variability in the boundary layer. The Olsen and Randerson (2004) method could be used to remediate part of this problem. However, this solution would not be as straightforward to apply in an online forecast as it is done in an offline mode, for which all the climate forcing parameters (2 m temperature and solar radiation can be retrieved beforehand). There are also other reasons for not using an offline NEE product or optimized fluxes directly in the CAMS $CO_2$ forecasting system:

- Downscaling the coarse optimized fluxes (2.5x3.75 degrees) at the resolution used by NWP models (currently 9 km at ECMWF) is not straightforward. Inconsistencies in the topography (particularly around mountains and coastlines) makes the low resolution fluxes difficult to use in a high resolution model.

- Coupling of $CO_2$ fluxes from terrestrial vegetation and the atmospheric model represents an important step towards a better understanding of the interaction between the ecosystem and regional atmospheric processes (Lu et al. 2001, Moreira et al, 2013). Boussetta et al. (2013) showed that the coupling between the $CO_2$ fluxes and the water and energy fluxes in the modelling of vegetation can improve the simulation of surface parameters such as temperature and humidity as well as NEE. This coupling has been shown to benefit the simulation of the $CO_2$ diurnal cycle in the atmospheric boundary layer in the tropics (Lu et al., 2005, Moreira et al. 2013).

- Finally, because offline NEE products or optimized fluxes are not available in near-real time, we would need to use a climatology. The inter-annual variability associated with the land sink cannot be considered when using just a climatology of NEE. Despite being a challenging aspect of the modelling, we think it is worth having inter-annual variabilily in the model forecast. The main rationale for this is based on the understanding that the climate variables simulated in the NWP model – such as temperature and precipitation – play an important role in explaining the inter-annual variability of NEE (Schaefer et al. 2002). The motivation for including the model inter-annual variability in the flux adjustment will be clarified in the revised manuscript.

**These three points above have been included in the Methodology section to explain the motivation behind the modelling of the CO2 fluxes online.**

* *It is not clear to me why certain metrics were chosen for evaluation. The authors present mean biases and standard deviations in Figures 9 and 10, correlation coefficients in Table 4, no metric for Figure 11, but there are never root- mean-square differences reported which I think are most useful. I think in figure 11 the MACC fluxes have the lowest RMSD than the BFAS fluxes. And from the captions it seems that both observations and simulations are done as daily (24-hour?) averages. I think that this daily averaging is needed because the independent adjustment of the GPP and TER scaling factors leads to strong variations in NEE that do not necessarily preserve a good diurnal cycle. But I might be wrong on that, as I could not assess this from the figures shown. 24-hour average observations could have a lot of hour-to-hour variability which should be shows by an error bar. The statistics and figures moreover seem to cover only the month of March and a few selected days in March. It remains unexplained why this choice was made, and what the metrics look like for other months. I would expect for instance in summer to see even larger day-to-day variations in NEE, and then also in atmospheric $CO_2$*

Following the reviewer's advice, we have computed the root-mean-square (RMS) error of the different $CO_2$ experiments with respect to observations at the tower sites shown in Fig 11 of the manuscript (see Table 1 below). With the RMS error it is not as easy to see the improvement in the modelled variability as with the correlation coefficient $r$, because the RMS error increases very rapidly when there is large variability. This effect can be clearly seen at Park Falls at 30 m above the surface. Despite the substantial improvement in the model variability with BFAS ($r = 0.8$) compared to the CONTROL forecast ($r = 0.3$), the RMS error is larger in BFAS than in the CONTROL experiment by more than 1 ppm. This happens because the BFAS experiment overestimates the amplitude of the synoptic variability which is nearly non existent or even anticorrelated in the CONTROL experiment. At West Branch, the BFAS experiment has a much lower RMS error than both the experiments without BFAS and with optimized fluxes. Table 1 can be included in the supplement of the revised manuscript.

**The RMS error has been included in the evaluation of the flux adjustment results in the revised paper (see Table 1 and Figs. 3 and 5 in the Supplement.)**

The impact of BFAS on the diurnal cycle amplitude has been evaluated in the northern hemisphere land (north of 20$^o$N) based hourly data from all the in situ stations compiled in the NOAA Obspack (2015) dataset for 2010 (Fig. 1 of this reply). The mean error of the diurnal cycle amplitude (daily max value minus daily min value) is reduced for all seasons, with larger improvements in winter, autumn and spring. The RMS error on the other hand is slightly worsened. This is not surprising since the reference optimized flux dataset is not designed to represent the synoptic variability of the diurnal cycle amplitude (see green and dark blue bars in Fig. 1 of this reply). Summer months have larger diurnal cycle amplitudes and as expected the model also has larger errors in JJA. However, the impact of BFAS on the RMS error is the same for all months.

**This assessment of the diurnal cycle has been included in the Supplement of the revised manuscript.**

* *I would like to know what the added value is of having the gamma-parameter included in BFAS. The description of its calculation and adjustment is quite extensive but I do not really understand what role it plays. Perhaps there could be an experiment where BFAS is used without the adjustment in equation 3. After all, not needing the ensemble of forecasts would make the scheme a bit simpler, and perhaps just as good? I know I am likely to be wrong as the authors have decided to include this procedure in BFAS, but I would like to see the evidence to support*

Table 1: Root mean square error [ppm] of different forecast (FC) experiments with observations at three NOAA/ESRL tall towers for daily mean dry molar fraction of atmospheric $CO_2$ in March 2010. The dash symbol means the correlation is not significant.

| NOAA/ESRL Tower site (ID) | Latitude, Longitude, Altitude | Sampling level [m] | BFAS FC | CTRL FC | OPT FC | OPT-CLIM FC |
|---|---|---|---|---|---|---|
| Park Falls, Wisconsin (LEF) | 45.95ºN, 90.27ºW, 472 m | 30 122 396 | 6.12 4.05 2.93 | 4.97 5.44 5.10 | 3.04 2.09 1.37 | 3.31 3.06 1.99 |
| West Branch, Iowa (WBI) | 41.72ºN, 91.35ºW, 242 m | 31 99 379 | 3.79 2.91 2.46 | 10.39 9.94 8.91 | 5.06 2.95 3.20 | 6.96 3.92 2.43 |
| Argyle, Maine (AMT) | 45.03ºN, 68.68ºW, 50 m | 12 30 107 | 3.72 3.55 2.86 | 3.76 3.36 3.37 | 2.35 1.66 1.06 | 1.30 0.82 0.76 |

*that decision.*

A new experiment has been performed in which the $\gamma$ factor is set to zero in order to demonstrate the value of having the inter-annual variability in BFAS. Indeed the inter-annual variability can be important factor in the simulation of $CO_2$ (Schaefer et al. 2002, Chamard et al. 2003). However, because is not the same in every region/season/year it can also be difficult to demostrate its impact with observations (Figs 2, 3, 4 and 5 of this reply). In BFAS, the use of the $\gamma$ factor to represent the inter-annual variability from the model generally has a small impact. However, there are seasons and regions where we see the impact of using the $\gamma$ factor. As expected, this impact tends to be larger in the tropics, where the model inter-annual variability is also largest (Agusti-Panareda et al. 2014). However, we can also see some impact in the northern and sourthern hemisphere for the MAM, JJA, SON seasons. In summary, including the inter-annual variability factor in BFAS is beneficial as in most cases it leads to a bias reduction, with just a few exceptions for the SON season (see LTrop in Fig. 4 and LN20N in Fig. 2 of this reply).

**The plots with the new experiment have been included in the Supplement. In addition, a summary of the experiment results has been included in the Methodology section 2.2 together with the rationale for including the inter-annual variability factor in the flux adjustment.**

**Minor comments**

*\* Page 3, line 5: I do not agree that the current monitoring of $CO_2$ relies on satellites and it is even a bit insulting to the real monitoring groups to say it. I suggest to change it because*

*satellites do not yet see reliable $CO_2$. In fact, the second part of this statement is also not right because the observations you show and that MACC fluxes rely on mostly come from flasks and not from in-situ instruments.*

The reference to in situ observations was meant to include both continuous and flask measurements (lines 8 to 10 in Page 3).

**In the revised version of the manuscript this has been clarified by specifying both explicitly.**

*\* Page 12, line 20: the current adjustment scheme for GPP and TER does not include any covariances between the adjustments, but we know that they often respond in the same direction and that errors are correlated. It would be good to think about an adjustment scheme that uses such information. Showing the posterior diurnal cycle is also needed.*

**This has been mentioned as future improvements planned for BFAS in section 6.3 of the revised manuscript. The impact on the diurnal cycle has been included in the supplement as mentioned above.**

*\* Page 13, line 20: You use now the names OPT-CLIM and later on in the text and tables CLIM-OPT. Is this the same run? It was to me confusing. Also see later remark about Table 2*

The runs are the same. **The text and Table 2 have been corrected in the revised version to use the consistent label for the OPT-CLIM experiment.**

*\* Page 14, line 20: A table listing the annual mean fluxes for transcom regions for all simulations would be valuable I think*

**The proposed table for the budget in the Transcom regions has been included in the Supplement of the revised manuscript.**

*\* Page 15, line 25: The SH problems could come from a different north to south transport characteristic of the two atmospheric models used (IFS and LMDZ?). Can this be illustrated with a simple SF6 simulation and compare it to observations?*

We think the negative bias in the southern hemisphere comes from biases in tropical Africa. Preliminary experiments to assimilate IASI $CO_2$ using the $CO_2$ forecast have shown a large systematic difference throughout the free tropospheric column over tropical Africa which is consistent with the negative bias in the southern hemisphere.

**This has been mentioned in the revised manuscript.**

*\* Acknowledgements: please check the data usage policy of NOAA as I do not believe you can simply take data from their FTP and then publish it with this statement.*

The authors have contacted Ed Dlugokencky regarding the acknowledgements and received his confirmation that these are sufficient. **An acknowledgement for the Obspack data used for the plots in the Supplement of the revised manuscript has been added.**

*\* Page 30, Table 2: I was confused because it says that CLIM-OPT uses MACC fluxes as reference in BFAS but from the methods I understood that CLIM-OPT or OPT-CLIM used the climatological fluxes from MACC directly as underlying biosphere fluxes? I discovered this only*

*towards the end of reading and it made me think I misunderstood the simulations completely. Even now I doubt it.*

CLIM-OPT uses the climatological fluxes from MACC (i.e. the total $CO_2$ flux) and BFAS justs uses a climatology of the MACC residual biosphere fluxes.

**This has been clarified in Table 2 and in the text of the revised manuscript.**

* *Figures 4 and 7: it would be better to use PgC/yr as units and not GtC/day because now they just look very small on the y-axis with many insignificant digits to start.*

If the units are changed to PgC/yr then the values have to be divided by 365, which result in even a larger number of insignificant decimal points. For this reason, the units have not been changed in the revised manuscript.

* *I believe Figure 12 and 13 are not needed and could be removed.*

The authors disagree on this point. The fact that BFAS can change the gradient of the fluxes and as a result improve the atmospheric $CO_2$ synoptic variability is an achievement that needs to be properly documented.

**Reply to reviewer 2**

**General comments**

• *This paper presents an enhancement to the CO2 assimilation system used within the Copernicus tracer assimilation system at ECMWF. The enhancement is certainly useful and potentially quite important but it comes with its own problems. I believe these need to be discussed in the manuscript and addressed in how the new product is made available.*

We thank the reviewer for his insightful comments concerning the potential use of the CAMS $CO_2$ analysis product in flux inversion systems. The reply to each of the reviewer's points can be found below, together with a pointer to the section of the text that has been modified in the revised manuscripts (see blue text). A highlight of all the modification introduced in the revised manuscript can be found in the latexdiff file provided.

• *The enhancement addresses the problem of large-scale biases in the fluxes which underlie the prior concentrations used in the assimilation. These biases are a serious matter since they mean that the probability densities assumed in the assimilation system (centered on the true value) dont, in fact, hold. So this is a potentially valuable improvement.*

This is a very important part of the motivation of this work because the atmospheric $CO_2$ forecast provides the prior information to the CAMS atmospheric $CO_2$ data assimilation. As the reviewer points out, the data assimilation system is only designed to reduce the random error, not the bias. Therefore, it is very important to bias correct the prior atmospheric mixing ratios from the forecast before assimilating any $CO_2$ observations. **We have included this point in the introduction of the revised manuscript to strengthen the motivation for BFAS.**

● *The problem arises when we consider what the generated CO2 fields are used for. Although there is probably some benefit for improved retrievals of temperature and moisture by improving the CO2 field the overwhelming use for the assimilated CO2 products is in estimating surface fluxes. the statistical apparatus is identical to the assimilation of the CO2 fields and the same restrictions apply. Among them is a firm prohibition on reusing information and the requirement that observations and prior are independent. Both of these are potentially violated in any downstream use of the BFAS product. Lets deal with these two problems in turn.*

The reviewer has an important point in that users of the CAMS $CO_2$ analysis/forecast products need to know what is the input data going into the product and what is the final uncertainty of the product. This is the case whether the users are working on flux inversion systems, planning of field experiments or using the product as boundary conditions for regional models.

Regarding the mixing of information in the analysis, this is currently not an issue for the CAMS $CO_2$ analysis system because the optimized fluxes used in BFAS are not based on satellite products; whereas the CAMS atmospheric $CO_2$ analysis is currently only assimilating satellite products.

For the users, we envisage that the atmospheric $CO_2$ analysis/forecast will be used as boundary conditions for regional flux inversion systems. In this case the possible correlated errors between such an analysis and the measurements assimilated by the inversion within the regional domain will likely be marginal, given all the processing that is involved between the inversion to estimate the MACC optimized fluxes, BFAS and the IFS 4D-Var used by the CAMS atmospheric $CO_2$ analysis. The possibility to infer the surface fluxes directly from the IFS $CO_2$ analysis would mean that some information from the observations assimilated by the MACC flux inversion system would already be present in the CAMS $CO_2$ analysis via BFAS. Thus, we have included information on the observations used in the flux inversion system to produce the optimised fluxes in the revised manuscript.

**Information on the uncertainty of the atmospheric $CO_2$ forecast with and without BFAS compared to the optimized flux experiments has been provided in terms of bias and root mean square error (RMSE) for different regions/seasons in the Supplement of the revised manuscript using barplots as shown in Figs 2, 3, 4 and 5 in this reply. All the issues relevant to users have been included in the Discussion as part of a new section entitled "Aspects to be considered by users".**

● *I believe this paper is a potentially valuable contribution and look forward to the authors revision. If the authors accept my first point about the mixing of data into their CO2 field then they also need to find a way of detailing which data was used to generate the flux fields that underlie BFAS.*

The flux fields underlying BFAS are primarily NEE modelled by the CTESSEL Carbon module in the IFS (Boussetta et al. 2013), which are then re-scaled using continental-scale climatological budgets from the MACC optimized fluxes of Chevallier et al. (2011, 2015). There is also some input from the EDGAR v4.2 anthropogenic emissions and the biomass burning emissions from GFAS (Kaiser et al. 2012). The information from these inventories is used to extract the NEE as a residual from the optimized fluxes.

**The documentation of the different data streams going into BFAS and their access (via the Copernicus Data Catalogue and the EDGAR database) has been included in the new Discussion subsection entitled"Aspects to be considered by users".)**

**Specific comments**

*• The assimilated CO2 field now includes information from a prior informed by a previous flux inversion. This inversion presumably used measurements from the in situ network, aircraft and/or TCCON. We cant tell which without a detailed examination of the papers that underlie that inversion. We need to know because, if were going to use the BFAS product to drive a future inversion, we need to exclude those measurements. One might argue that the periods dont overlap but the evidence of the paper shows that the model-data mismatch is so strongly correlated from year to year (consistent seasonal errors in the pre-BFAS version) that this doesnt avoid the problem.*

**In the revised manuscript we have mentioned that since the BFAS product contains information from the optimized fluxes, users should be aware that the optimized fluxes assimilated most available background air-sample monitoring sites (listed in the supplement of Chevallier et al. 2015, see `http://www.atmos-chem-phys.net/15/11133/2015/acp-15-11133-2015-supplement.pdf` (see section 6.4 in the revised manuscript).**

Although we expect that observations ingested by the MACC inversion system of Chevallier et al (2011, 2015) will have an influence on the BFAS fluxes to some extent, we cannot quantify their degree of influence in this paper. We expect some information from the observations will be lost in the flux inversion process and specially in BFAS. The processing in BFAS involves spatial/temporal smoothing of the optimized fluxes over land with a 10-year averaging to construct the climatology and then the inclusion of the model interannual variability. The influence from these surface observations will be further diminished after the assimilation of satellite products in the analysis. In order to ensure independence between the $CO_2$ analysis and the background-air observations ingested by the MACC inversion system, the atmospheric $CO_2$ analysis could be sampled at non-background-air locations characterized by a large influence from the satellite products.

*• The second problem, of the prior estimate for a flux inversion being partially reflected in the data we use is not new with BFAS. It exists in the original Copernicus products too. Im unsure whether the mixing data and model information in the prior CO2 field makes this problem worse but it seems like it should.*

The BFAS processing should bring the mean error and large-scale spatial distribution of the CTESSEL NEE fluxes closer to the MACC optimized fluxes. This probably implies that the BFAS fluxes will not be completely independent from the prior in the MACC flux inversion system. Thus, if the same prior would be used again to infer fluxes from the atmospheric $CO_2$ analysis data, then it would be likely that BFAS would make the problem associated with their lack of independence worse.

*• Finally there is the question of the uncertainty of the BFAS CO2 field. There are two countervaling effects in play. First the bias correction of the prior has reduced residuals in the generated CO2 field so that uncertainties (which are the statistics of the differ- ence between estimated and true values) seem to have reduced. On the other hand an extra process has been added to the assimilation with a new set of parameters to scale prior fluxes. These will have their own uncertainty and should (since the posterior CO2 field is sensitive to its prior) increase posterior uncertainty. Which of these wins out? I am always a little wary of criticizing a paper for things it did not do since no piece of research is complete. However its an important general rule that*

*products that are to be used as inputs to statistical procedures such as flux inversions need to specify their uncertainty as well as their mean.*

**Plots showing characteristic biases and root mean square errors of the BFAS $CO_2$ field for different seasons/regions have been included in the supplement of manuscript (see Figures below) and referred to in the new section 6.4 entitled "Aspects to be considered by users". These plots use all the observations from the NOAA Obspack (2015) dataset (excluding only the observations from CONTRAIL and HIPPO flights).**

[Figure]

*Figure 1: Evaluation of diurnal cycle amplitude of $CO_2$ dry molar mixing ratio [ppm] for the different forecast experiments (see legend) in the northern hemisphere land (north of $20^oN$) based on hourly data from all the in situ stations compiled in the NOAA Obspack (2015) dataset for 2010. Top panel: mean error; middle panel: root mean square error; and lower panel: number of observations.*

[Figure]

Figure 2: Mean error of atmospheric $CO_2$ dry molar mixing ratio [ppm] for different forecast experiments (see legend) with respect to insitu and flask observations for different seasons and regions (N20N: north of $20^oN$; Trop: between $20^oS$ and $20^oN$; S20S : south of $20^oS$) with a separation between land and sea points denoted by a preceeding "L" and "S" in the region name respectively. The observations were extracted from the NOAA Obspack (2015) dataset in 2010. The number of observations used for the statistics are shown as grey bars in the panel below each plot.

[Figure]

Figure 3: Root mean square error of atmospheric $CO_2$ dry molar mixing ratio [ppm] for different experiments (see legend) with respect to insitu and flask observations for different seasons and regions as described in Fig. 2. The observations were extracted from the NOAA Obspack (2015) dataset in 2010. The number of observations used for the statistics are shown as grey bars in the panel below each plot.

[Figure]

Figure 4: Mean error of atmospheric $CO_2$ dry molar mixing ratio [ppm] for different experiments (see legend) with respect to NOAA aircraft vertical profiles (Sweeney et al. 2015) in the free troposphere (1000 m above surface) for different seasons and regions as described in Fig. 2. The observations were extracted from the NOAA Obspack (2015) dataset in 2010. The number of observations used for the statistics are shown as grey bars in the panel below each plot.

[Figure]

Figure 5: Root mean square error of atmospheric $CO_2$ dry molar mixing ratio [ppm] for different experiments (see legend) with respect to NOAA aircraft vertical profiles (Sweeney et al. 2015) in the free troposphere (1000 m above surface) for different seasons and regions as described in Fig. 2. The observations were extracted from the NOAA Obspack (2015) dataset in 2010. The number of observations used for the statistics are shown as grey bars.

The $\gamma$ inter-annual variability factor is multiplied by the standard deviation of the optimised residual NEE budget – representing the typical amplitude of inter-annual variability – in order to offset the reference climatological NEE budget. In this way, the inter-annual variability of the reference NEE follows the inter-annual variability of the model NEE with the same anomaly sign, while keeping its amplitude constrained by the standard deviation of the optimised flux budget. Note that the use of this factor is optional. By setting it to zero, the model budget can be constrained by the optimized flux climatology. The rationale for applying this factor in the C-IFS system is based on the fact that inter-annual variability of the NEE budget is strongly linked to the inter-annual variability of climate variables such as precipitation and temperature (Schaefer et. al., 2002). Since information on these climate variables is readily available in the C-IFS system, it is worth exploring its impact on the $CO_2$ forecast. A preliminary assessment of the impact of including the inter-annual variability factor was performed by comparing experiment with and without the factor. Results confirmed a small but positive impact (see Supplement). Details on the computation of this factor are given in the next section.

**2.3 The inter-annual variability factor**

The computation of the inter-annual variability factor $\gamma$ requires a model climate consistent with the forecast (i.e. same meteorological analysis, same model version and same resolution). Producing a consistent model climate is not a trivial requirement, because both the operational model version and analysis system can change frequently with new updates and new observations, and high resolution forecasts spanning a period of 10 years (i.e. 2004 to 2013) are expensive. A feasible solution has been found where the standardised NEE anomaly from the model is computed using the operational Ensemble Prediction System (ENS) forecasts and hindcasts which are part of the ECMWF monthly forecasting system (Vitart et al., 2008; Vitart, 2013, 2014). Every Monday and Thursday the operational ENS is not only run for the actual date, but also for the same calendar day of the past 20 years. These hindcasts have the same resolution and model version as the ENS forecasts and they constitute a valuable data set used for the post-processing and calibration of the NWP forecasts from the medium-range (10 days) up to one month lead times (Hagedorn et al., 2012). The ensemble of forecasts is made of 5 members (10 members since 2015) using perturbed initial conditions (Lang et al., 2015) and stochastic physics in order to represent forecast uncertainty (Palmer et al., 2009).

As the hindcasts are not performed daily, it is not possible to aggregate consecutive 1-day forecasts into a 10-day period to compute a mean budget as shown in Fig. 2. In order to circumvent this, the mean budget is computed by averaging the 1-day forecast NEE from all the ensemble members available in the hindcasts. This is done for each year from 2004 to 2013 to preserve consistency with the NEE climatology from the optimised fluxes. The model climate $f^{\mathrm{Mclim}}$ given by the 10-year mean budget and its typical inter-annual variability $\sigma\left(f^{\mathrm{Mclim}}\right)$ can then be obtained by calculating the mean value and standard deviation respectively over that period. Similarly, the model budget $f^{\mathrm{M}}$ is calculated from the NEE ensemble mean of the ENS forecast for the current date using the same number of ensemble members as the ENS hindcasts. The standardised anomaly $\gamma$ is finally obtained by subtracting the 10-year mean budget from the current budget and dividing the anomaly by the standard deviation. Since the hindcasts are available every Monday and Thursday, $\gamma$ is only updated twice a week. These updates are routinely monitored during the forecast (see Sect. 4).

**2.4 Partition of NEE adjustment**

The final stage in the flux adjustment is the attribution of the NEE correction to the different biogenic fluxes in the model. The residual NEE from optimised fluxes only provides information on the total flux from the land ecosystem exchange. While in land vegetation models, NEE is the combination of two opposing fluxes: Gross Primary Production (GPP) and the ecosystem respiration ($R_{\mathrm{eco}}$). Given that we have no information on whether the NEE error is associated with the GPP or the $R_{\mathrm{eco}}$ fluxes, a strategy has to be defined in order to partition the NEE correction into GPP and $R_{\mathrm{eco}}$. The underlying strategy used here is to have the smallest flux adjustment possible. Namely, the scaling factors should be as close to 1 as possible.

The first step is to distinguish between the positive and negative values of the NEE scaling factor ($\alpha$). A positive NEE scaling factor implies the budget of the NEE in the model has the correct sign but the wrong magnitude. In that case, the scaling of the flux will be smallest if the dominant component of NEE is scaled. That is to say, the flux correction will be applied to GPP during the growing season and to $R_{\mathrm{eco}}$ during the senescence period. Whereas if the scaling factor is negative – i.e. the modelled NEE has the wrong sign – only the flux with smallest magnitude is corrected (GPP or $R_{\mathrm{eco}}$) to ensure the scaling factor of the modelled fluxes is always positive.

The scaling factor $\alpha$ is then converted into a scaling factor for the dominant component of the NEE flux. If the magnitude of GPP is larger than the magnitude of $R_{\mathrm{eco}}$, then the scaling factor for GPP and $R_{\mathrm{eco}}$ are defined as follows:

$$\alpha_{\mathrm{GPP}} = \frac{\alpha\mathrm{NEE} - R_{\mathrm{eco}}}{\mathrm{GPP}}$$
$$\alpha_{R_{\mathrm{eco}}} = 1.0 \tag{4}$$

Similarly, if $|R_{eco}| > |GPP|$ then

$$\alpha_{GPP} = 1.0$$

$$\alpha_{R_{eco}} = \frac{\alpha NEE - GPP}{R_{eco}} \qquad (5)$$

[revised manuscript text omitted]